# Clumped-isotope-derived climate trends leading up to the end-Cretaceous mass extinction in northwest Europe

Heidi E. O'Hora[1], Sierra V. Petersen[1], Johan Vellekoop[2,3], Matthew M. Jones[1,4], Serena R. Scholz[1]

[1]Department of Earth and Environmental Sciences, University of Michigan, Ann Arbor, MI 48104, USA
[2]Analytical, Environmental, and Geo-Chemistry, Vrije Universiteit Brussel, B-1050 Brussels, Belgium
[3]Department of Earth and Environmental Sciences, KU Leuven, 3000 Leuven, Belgium
[4]Smithsonian Institute, Washington, D.C., 20013, USA

*Correspondence to*: Sierra V. Petersen (sierravp@umich.edu)

**Abstract.** Paleotemperature reconstructions linked to Deccan traps volcanic greenhouse gas emissions and associated feedbacks in the lead-up to the end-Cretaceous meteorite impact and extinction document local and global climate trends during a key interval of geologic history. Here, we present a new clumped-isotope-based paleotemperature record derived from fossil bivalves from the Maastrichtian type region, in southeast Netherlands and northeast Belgium. Clumped isotope data documents a mean temperature of 20.4 ± 3.8 °C, consistent with other Maastrichtian temperature estimates, and an average
seawater $\delta^{18}O$ value of 0.2 ± 0.8‰ VSMOW for the region during the latest Cretaceous (67.1–66.0 Ma). A notable temperature increase at ~66.4 Ma is interpreted to be a regional manifestation of the globally-defined Late Maastrichtian Warming Event, linking Deccan Traps volcanic $CO_2$ emissions to climate change in the Maastricht region. Fluctuating seawater $\delta^{18}O$ values coinciding with temperature changes suggest alternating influences of warm, salty southern-sourced waters and cooler, fresher northern-sourced waters from the Arctic Ocean. This new paleotemperature record contributes to the understanding of regional
and global climate response to large-scale volcanism and ocean circulation changes leading up to a catastrophic mass extinction.

## 1 Introduction

During the late Maastrichtian, greenhouse gas emissions from the emplacement of the vast Deccan Traps large igneous province (LIP) on the Indian subcontinent resulted in global warming (see Hull et al., 2020 and references therein). This
warming event (termed the Late Maastrichtian Warming Event or LMWE; Woelders et al., 2018) has been observed in multiple locations and has been dated to approximately coincide with the onset of major Deccan volcanism (66.413 ± 0.067 Ma; Sprain et al., 2019). Anomalous mercury concentrations in sediments (e.g., Font et al., 2016; Sial et al., 2016; Percival et al., 2018; Zhao et al., 2021) and fossil shells (Meyer et al., 2019) link the warming to a volcanic source, while modeling of the climate impacts of hypothesized volcanic $CO_2$ and Hg emissions result in warming consistent with observations (Tobin et al., 2017;
Fendley et al., 2019; Nava et al., 2021).

Prior to the LMWE, global climate was in a "cool greenhouse" state (Scotese, 2021), with climate models (Miller et al., 2005; Ladant and Donnadieu, 2016), sea-ice-indicating dinoflagellate cysts (Bowman et al., 2013), and coastal Antarctic temperatures near the freezing point (Petersen et al., 2016a) indicating the possible existence of land ice on the Antarctic

continent. The LMWE may have caused such an ice cap to melt (Bowman et al., 2013; Petersen et al, 2016a) and, subsequently, global sea level to rise. This is potentially visible in the global sea level curve of Haq (2014), but the resolution of this sea level record is too coarse to confidently link cycles to the short-term LMWE.

After the Late Maastrictian Warming Event, climate cooled again gradually, potentially due to silicate weathering feedbacks

(Petersen et al., 2016a; Tobin et al., 2017), before being thrown into upheaval by the arrival of the Chicxulub meteorite impact, defining the Cretaceous-Paleogene (K-Pg) boundary (Molina et al., 2006). Ejecta from the impact covered the planet and resulted in a short-lived "impact winter" event (Vellekoop et al., 2014; Vellekoop et al., 2016). Taken together, Deccan Traps volcanism and associated feedbacks and the Chicxulub bolide impact create a complex pattern of paleoenvironmental disturbances through the end-Cretaceous, ultimately leading to the rapid extinction of ~70 % of species on Earth (Schulte et

al., 2010).

Previous studies have reconstructed Maastrichtian marine temperatures at sites around the globe using several different proxies, including $\delta^{18}O$ of foraminifera, molluscs, tooth enamel, and bulk carbonate material (e.g., Pucéat et al., 2007; Tobin et al., 2012; Hull et al., 2020); $TEX_{86}$ measurements (e.g., Vellekoop et al., 2016; Woelders et al., 2017); Mg/Ca ratios (e.g., Woelders

et al., 2018); and clumped isotope thermometry (Dennis et al., 2013; Tobin et al., 2014; Petersen et al., 2016a; Petersen et al., 2016b, Meyer et al., 2018; Zhang et al., 2018; Meyer et al., 2019; Tagliavento et al., 2019) (Table S1). Taken together, these records clearly show a late Maastrichtian warming (summarized by Hull et al., 2020), although the magnitude of warming differs from site to site. At any given site, ocean temperatures across the LMWE may be influenced by the primary $CO_2$-driven global warming, which may have manifested with amplified warming at high latitudes similar to modern Arctic amplification

(Petersen et al., 2016a), and also by local changes in ocean circulation, sea level, upwelling, runoff, etc. (changes which may themselves by driven indirectly by global climate change, e.g., sea level rise due to melting of the Antarctic ice caps during the LMWE). Therefore, it is important to consider the local paleogeographic context when interpreting paleoclimate trends during this tumultuous period.

Here, we produce a new climate record for a site in northwest Europe using the clumped and stable isotopic composition of bivalve fossils to better constrain the magnitude of climate change and possible ocean circulation fluctuations during this key geologic transition interval. We focus on the Maastrichtian type area (Jagt and Jagt-Yazykova, 2012) in the Belgium-Netherlands border region near Maastricht, Netherlands. The reconstructed paleogeography of this locality—a shallow epicontinental sea surrounded by several low-lying landmasses (Fig. 1a)—makes this site sensitive, not only to global climate

changes, but to local variations in sea level and ocean circulation. We interpret our data in terms of previously described and modeled estimates for local, regional, and global climate changes during this time. Our study adds to a growing body of literature documenting the impacts of volcanism and ocean circulation on climate, especially leading up to the K-Pg extinction.

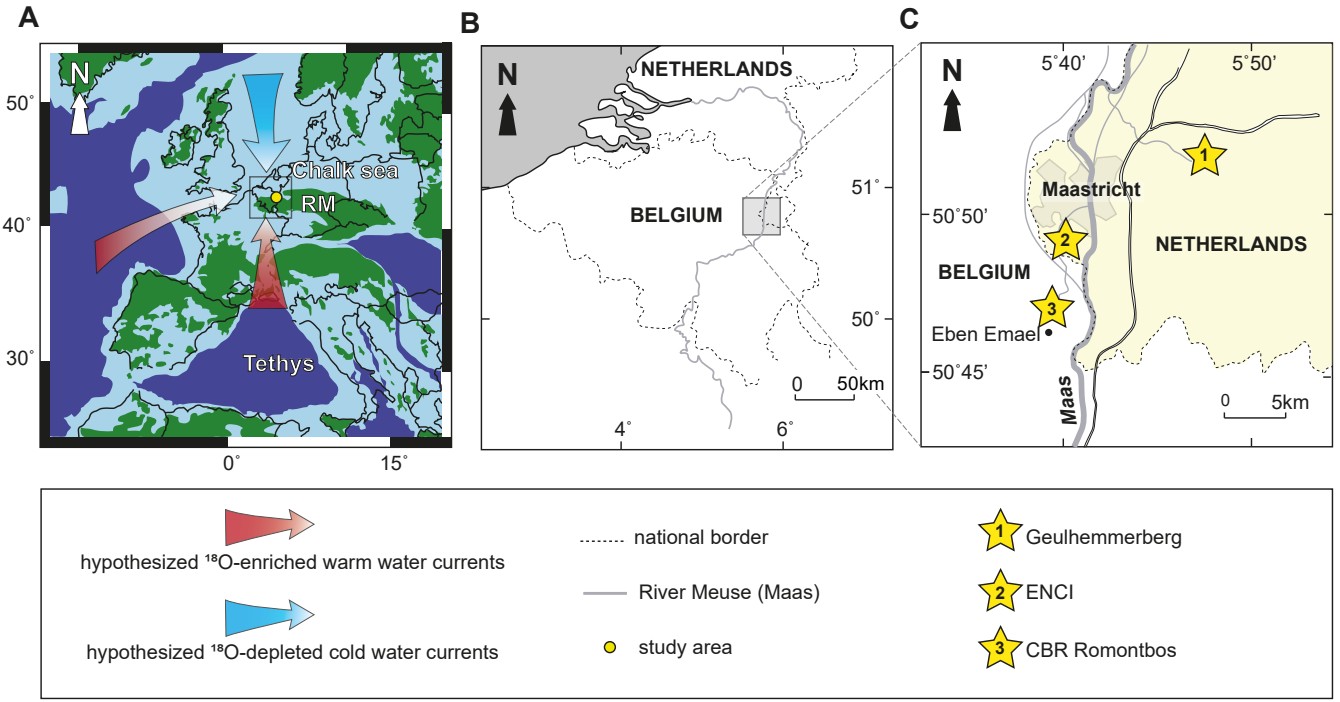

**Figure 1: Paleogeography and sample localities used in this study.** (a) Late Cretaceous paleogeographic reconstruction of Europe and surrounding regions based on Vellekoop et al. (2022) with the study area (yellow dot) and hypothesized cold and warm water currents (blue and red arrows) marked. (b) A map of modern national borders showing the position of the study area. (c) A zoomed-in map of the Maastricht region, showing the location of the ENCI Quarry, CBR Romontbos quarry, and Geulhemmerberg subterranean galleries relative to each other and to the city center of Maastricht.

## 2 Locality and Sample Collection

Shell specimens were collected from three locations in the Liège-Maastricht region of northeastern Belgium and southeastern Netherlands: the Maastrichtian type section at ENCI quarry, the CBR Romontbos quarry, and the K-Pg boundary (KPB) section at the Geulhemmerberg subterranean galleries (GSGs) (Jagt and Jagt-Yazykova, 2012) (Fig. 1c). The ENCI quarry is located ~3.3 km south of the city of Maastricht in South Limburg, Netherlands (Fig. 1c). Maastrichtian-aged strata at this location include the Gulpen Formation (~67.1–78.9 Ma) overlain by the Maastricht Formation (~66.0–66.8 Ma) (Vellekoop et al., 2022.), the latter of which was sampled for this study of the uppermost Maastrichtian Stage. At this location, the stratigraphically highest exposure of the Maastricht Formation in the Meerssen Member predates the KPB by less than 100

kyr (Keutgen, 2018). The CBR Romontbos quarry is located ~4 km to the southwest of ENCI quarry, near Eben Emael, Liège, Belgium (Fig. 1c). The Maastricht Formation is also exposed at this location, and samples from here were correlated to align

with the Maastricht Formation at ENCI quarry (Felder et al., 1975). We also include a specimen from the nearby Geulhemmerberg subterranean galleries, located ~7 km east-northeast of ENCI quarry (Fig. 1c). The exposed section in the GSGs lies stratigraphically above the ENCI quarry and preserves the uppermost section of the Meerssen Member, including the KPB itself (Felder et al., 1998; Keutgen et al., 2018). Between the top of the ENCI section and the short KPB interval recorded in the GSGs, a time gap exists of no more than several tens of thousands of years (Vellekoop et al., 2022).

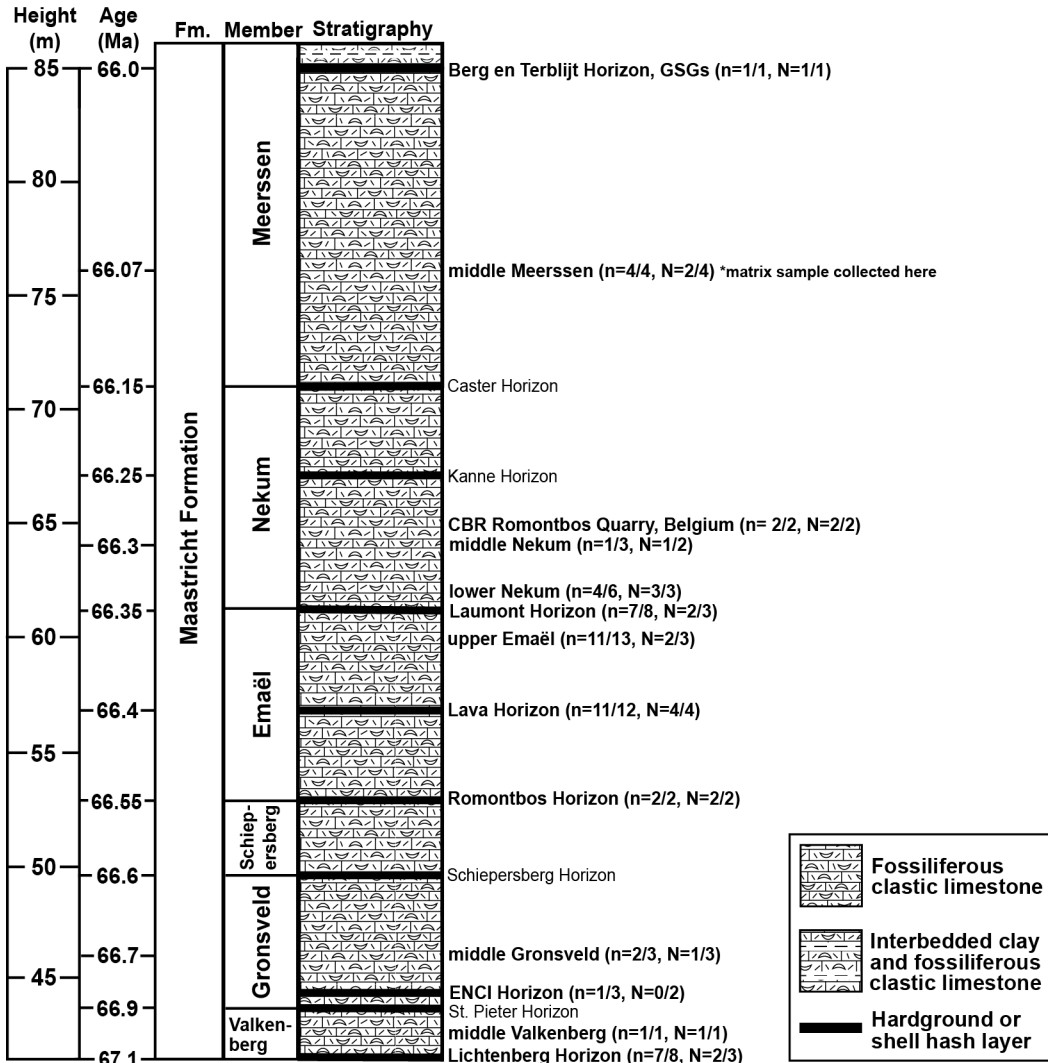

**Figure 2: Composite stratigraphic section through the Maastricht Formation at ENCI quarry, Romontbos quarry, and Geulhemmerberg subterranean galleries.** Samples collected in the Romontbos quarry and GSGs were placed on the ENCI quarry stratigraphic framework using the age model from Vellekoop et al. (2022) (Table S2). From each collection horizon or zone, the number of specimens collected, the number that passed diagenesis screening, the number of well-preserved specimens successfully analyzed for clumped isotopes and the total number attempted for clumped isotopes (including some


poorly-preserved) are listed as follows (n=passed/collected, N=good clumped/attempted). Out of 66 shells sample, 54 shells passed screening, 33 were attempted, and 27 well-preserved shells were successful for $\Delta_{47}$, plus one sample of matrix material from the Meerssen Member.

At ENCI quarry and in the surrounding region, the Maastricht Formation contains multiple conspicuous fossiliferous horizons (Felder, 1975; Zijlstra, 1988; Schiøler, 1997), including nine horizons in the Maastricht Formation at ENCI quarry (Fig. 2). Fossil shell specimens were collected from five of these horizons (Lichtenberg, ENCI, Romontbos, Lava, and Laumont), as well as from the top of the Emaël Member, the basal and middle intervals of the Nekum Member, and the basal interval of the Meerssen Member (Fig. 2). Ages for all stratigraphic horizons, lithostratigraphic boundaries, and specimens were determined

by updating the Keutgen (2018) age model using known biostratigraphic markers and bulk carbon isotope data (Vellekoop et al., 2022) (Table S2). Most horizon/unit ages have an uncertainty of ± 0.05 Ma, with the exception of the Lichtenberg Horizon (uncertainty of ± 0.2 Ma), Valkenburg Member and St. Pieter Horizon (both ± 0.1 Ma) (Table S2). Two samples were collected from the middle Nekum Member at the CBR Romontbos quarry, from an interval correlated to be above the middle Nekum Member samples from ENCI but below the Kanne Horizon.


The K-Pg boundary is positively identified in the Maastricht region, and in particular in the GSGs, through several lines of evidence (Smit and Brinkhuis, 1996; Vellekoop et al., 2020). The typical planktonic foraminiferal "disaster" assemblage (Smit and Zachariasse, 1996), the presence of the earliest Paleocene dinocyst marker taxon *Senoniasphaera inornata* (Brinkhuis and Schiøler, 1996; Herngreen et al. 1998), and $^{87}Sr/^{86}Sr$ analyses of well-preserved foraminifera from the clay layers of the

Geulhemmerberg underground galleries (Vonhof and Smit 1996) have all demonstrated that the K-Pg boundary occurs at the base of IVf-7 of the Meerssen Member, at the hardground known as the Berg en Terblijt Horizon (Fig. 2). Directly above this horizon, a sequence of interbedded layers of shell hash and clays represent a period of unique deposition, interpreted to represent a series of high energy events (storms or seismites) in the 100–300 years immediately following the Chicxulub meteorite impact (Roep and Smit, 1996; Smit and Brinkhuis, 1996; Vellekoop et al., 2020). The one oyster sample collected

from these shell hash layers was inferred to have been alive either within this short interval following the KPB, or be reworked from just underneath the Berg en Terblijt Horizon, and was thus assigned an age of 66.0 Ma (Clyde et al., 2016; Sprain et al., 2018) with an uncertainty of ± 0.01 Ma (Vellekoop et al., 2022), equivalent to the KPB at the scale of our time series (Table S2).

Fossil shell specimens collected from the Maastricht Formation at the ENCI quarry, the CBR Romontbos quarry, and the GSGs included 66 bivalve specimens representing eleven identified taxa of mostly oysters, with a few scallops and clams (from most to least common: *Acutostrea uncinella, Agerostrea ungulata, Pycnodonte vesicularis, Gryphaeostrea canaliculate, Entolium membranaceum, Neithea regularis*, *Pinna* sp.*, Pseudoptera* sp.*, Ceratostreon* sp.*,* and *Rastellum* sp. and indet. Ostreida). Six of the samples were only identified as some type of bivalve or oyster. The majority (40 of 66) are either *Acutostrea uncinella*

or *Agerostrea ungulata*. All identified specimens were sessile epifaunal suspension feeders and are assumed to have avoided

significant lateral postmortem transport. In these units, all biogenic aragonite has dissolved, leaving behind hollowed molds. All taxa sampled here originally formed calcite shells which continue to be preserved as calcite today (Fig. 3). Water depths for this location are estimated to be less than 20 m at all times, and often shallower (Hart et al., 2016), so water temperatures reconstructed from bottom-dwelling mollusks are reasonably interpreted to approximate mean annual surface air temperatures.


One sample of carbonate-rich matrix material from the Meerssen Member was analyzed for comparison to fossil shells. The carbonate in this specimen represents a combination of calcitic microfossils and possibly secondary, diagenetic calcite. As such, comparison between its isotopic and trace element composition and that of fossil shells can help assess the potential for diagenetic alteration of shell samples.

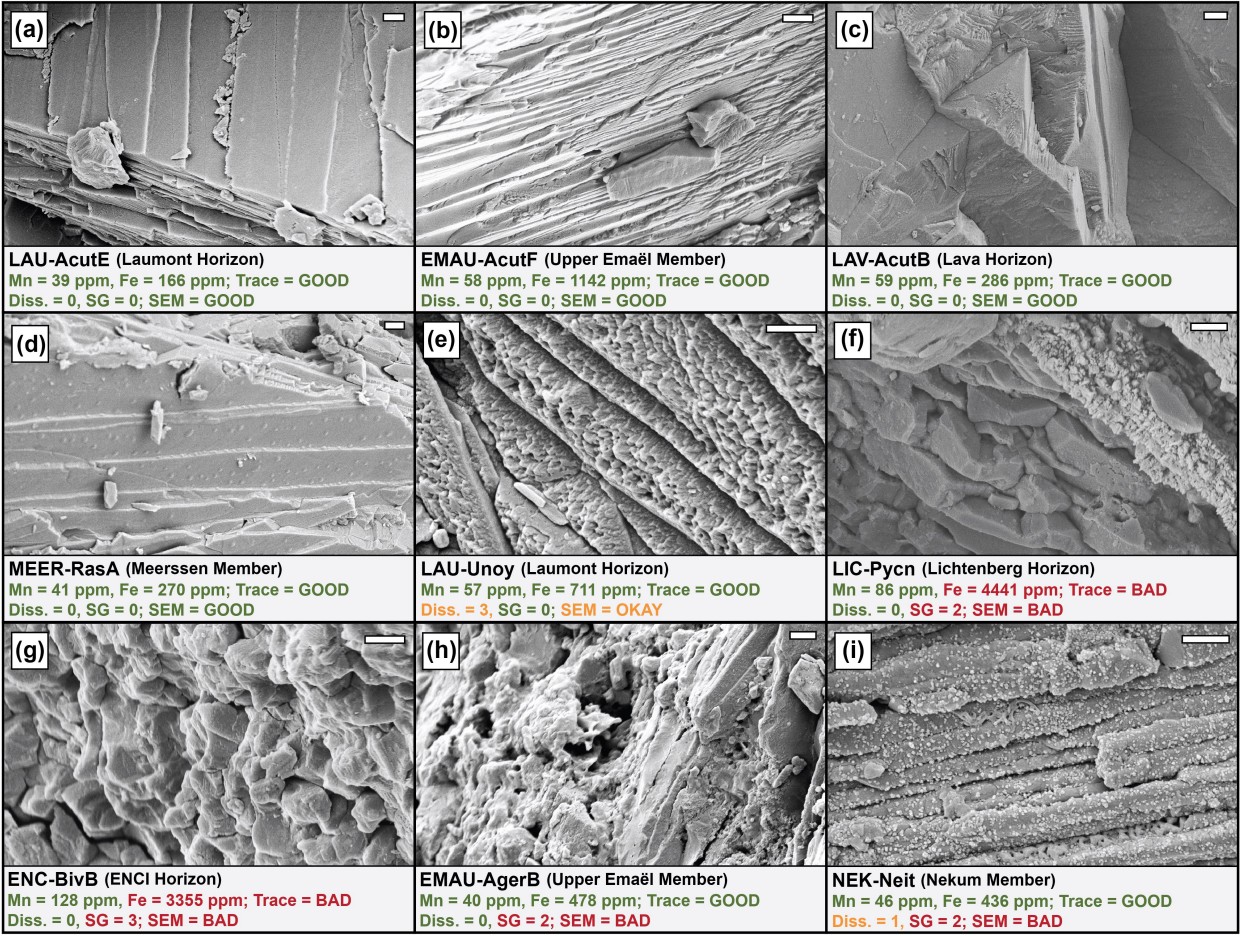


**Figure 3. SEM images of nine shell specimens representing varying levels of calcite preservation**. Scale bars represent 1 μm. Images were adjusted for brightness in Adobe Photoshop 2022. Trace element concentrations (in ppm) and SEM preservation indices quantifying the level and prevalence of dissolution and secondary growth (SG) on a scale of 0–3 are shown below each image for comparison. Samples (a) through (e) were interpreted. Samples (f) through (i) were not interpreted further, due to too much evidence of secondary growth and/or trace elements exceeding thresholds (see Fig. 4).


## 3 Methods

### 3.1 Sample preparation

After removal of matrix material through dionized water rinses and scrubbing with a toothbrush, fossil shell specimens were
sampled by either hand-drilling powder from larger fragments or manually crushing smaller fragments in their entirety in a
mortar and pestle. The one sample of matrix material from the Meerssen Member was scraped off a fossil and powdered in a
mortar and pestle to achieve roughly uniform grain size. The sample powders were divided for trace element analysis and
isotopic analysis. Fragments of some shell specimens were mounted for SEM observation to assess their preservation state
(Fig. 3).

Taxa-specific clumped isotope vital effects in bivalves are considered uncommon, based on studies of modern bivalve taxa
(Eagle et al., 2013; Henkes et al., 2013; Came et al., 2014), although one recent study documented apparent vital effects in the
juvenile portion only of the oyster *Magellana gigas* (Huyghe et al., 2022). In shells sampled for this study, we estimate that
juvenile shell material made up a volumetrically small fraction of sampled shell areas. Combined with the fact that our selected
species differ from the taxa demonstrating vital effects (Huyghe et al. 2022), we interpret well-preserved specimens as faithful
recorders of paleoenvironmental temperatures at the time of life.

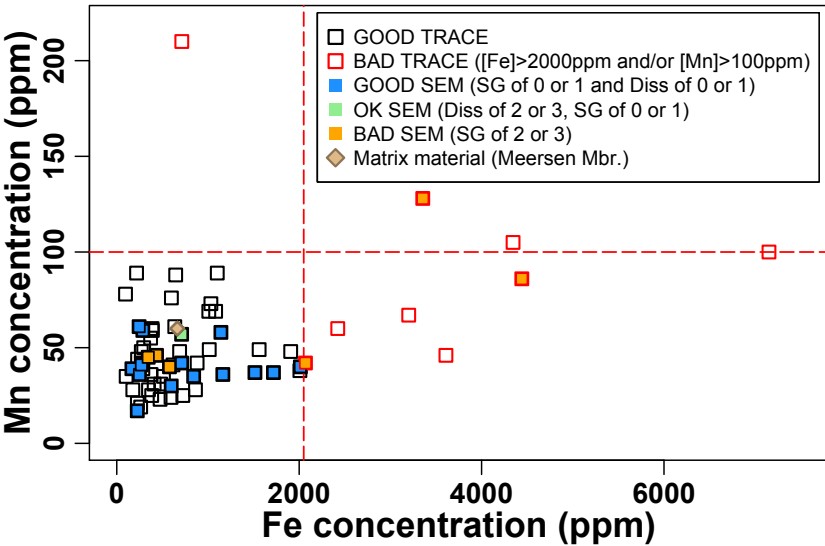

**Figure 4: Iron and manganese concentrations of all samples, used for diagenesis screening.** Samples with Fe
concentrations above 2500 ppm or Mn concentrations above 100 ppm, or both, were not considered for further interpretation.
These thresholds were chosen based on prior studies (see text), then adjusted to lower, more conservative levels to screen out
most samples showing secondary growth while keeping all samples showing good preservation or solely dissolution under
SEM (see Fig. 4).

## 3.2 Preservation assessment (SEM and trace elements)

The isotopic composition of fossil shells is susceptible to post-depositional alteration via recrystallization, which can skew interpreted formation temperatures (Brand and Morrison 1987). Elevated levels of trace elements (Mn and Fe especially) in calcitic fossil material may indicate diagenesis/recrystallization of the calcite (Land, 1967; Möller and Kubanek, 1976; Brand and Morrison, 1987; de Winter et al., 2018). Additionally, the prevalence of diagenetic calcite overgrowth on a sample can be identified easily under scanning electron microscopy (SEM) (Fig. 3), as diagenetic calcite does not show the same

microstructures as molluscan calcite. All but one of the fossil shells collected for this study were screened for recrystallization using a combination of SEM imaging and trace element detection methods (Fig. 3, Fig. 4, Table 1, Table S3).

All but one of the 66 shells (as well as one matrix sample) were analyzed in the Michigan Elemental Analysis Laboratory at the University of Michigan for Mn and Fe concentrations (Fig. 3) using a Thermo Scientific Quadrupole-ICP-MS (iCAP Q)

equipped with an Elemental Scientific PrepFAST 2 Automation System. KED mode (He gas) was utilized for all isotopes. All samples were analyzed in 2 % $HNO_3$ (w/v) (Optima grade), and sensitivity drift was corrected for with standard-sample bracketing methods (Table S4).

A subset of 21 specimens representing most horizons were photographed under SEM using a JEOL JSM-7800FLV field-

emission SEM in the Robert B. Mitchell Electron Microbeam Analysis Lab at the University of Michigan Ann Arbor (Fig. 3). SEM images were used to assign samples a score of 0–3 for both prevalence of dissolution (Diss) and secondary growth (SG) (Table S3). Samples that scored a two or three for SG were deemed "bad" and not considered for isotopic interpretation as the addition of diagenetic calcite with a (presumably) different isotopic composition than the base shell would skew environmental interpretations. Samples with some evidence of dissolution (score of two or three) but no secondary growth were deemed

"okay" and were still considered further because the material that remained likely still represented inhabited environmental conditions. Samples with scores of zero or one for both SG and dissolution were deemed "good" and were fully considered.

## 3.3 Stable and clumped isotope analysis

Powders drilled or crushed from all shell specimens and one matrix sample were analyzed for oxygen ($\delta^{18}O$) and carbon ($\delta^{13}C$) stable isotope compositions (Fig. S1, Fig. S2, Fig. S3, Table 1). Powders from 39 shells were analyzed using a Thermo-

Finnigan Kiel IV automated carbonate device coupled to a MAT 253 gas source dual inlet mass spectrometer in the University of Michigan Stable Isotope Laboratory. Data were corrected for acid fractionation and normalized to the VPDB scale using a single-step transfer function between two anchor points, the NBS-18 and NBS-19 standards. Precision was maintained at less than 0.1 ‰ for both $\delta^{18}O$ and $\delta^{13}C$ (Table 1, Table S4).


**Table 1: All isotope results.** Sample names created following a convention of 3–4 capital letters relating to the horizon, then 3–4 letters relating to the taxa, then one capital letter to separate multiple shells of the same taxa from that horizon. Age model from Vellekoop et al. (2022) (see Table S2). The preservation index (P. index) column combines the results of trace element analysis (GT = good trace, flag for either Fe, Mn, both (FeMn), or NA if no measurement) and SEM imagery (GOOD, OK, BAD, or NA if not imaged) in the format of trace/SEM. Any sample that failed $\geq 1$ criteria was not considered further. Errors for $\Delta_{47}$, temperature, and $\delta^{18}O_{sw}$ are external 1SE (see text).

| Sample Name | Age (Ma) | Height (m) | P. Index | $\delta^{18}O$ | $\delta^{18}O$ SD | $\delta^{13}C$ | $\delta^{13}C$ SD | $\Delta_{47}$ | $\Delta_{47}$ SE | Temp (°C) | Temp SE (°C) | $\delta^{18}O_{sw}$ | $\delta^{18}O_{sw}$ SE |
|---|---|---|---|---|---|---|---|---|---|---|---|---|---|
| LIC-Psop | 67.1 | 41.45 | GT/NA | -1.3 | 0.04 | 2.4 | 0.02 | - | - | - | - | - | - |
| LIC-Pycn | 67.1 | 41.45 | Fe/BAD | -1.3 | 0.1 | 2.2 | 0.1 | 0.698 | 0.009 | 22.0 | 2.9 | 0.4 | 0.6 |
| LIC-AcutA | 67.1 | 41.45 | GT/GOOD | -1.2 | 0.1 | 2.7 | 0.1 | 0.699 | 0.011 | 21.7 | 3.6 | 0.5 | 0.8 |
| LIC-Gry | 67.1 | 41.45 | GT/NA | -1.2 | 0.2 | 2.5 | 0.1 | 0.695 | 0.013 | 23.1 | 4.4 | 0.7 | 0.9 |
| LIC-AcutB | 67.1 | 41.45 | GT/NA | -1.1 | 0.1 | 2.9 | 0.03 | - | - | - | - | - | - |
| LIC-Ager | 67.1 | 41.45 | GT/NA | -1.0 | 0.02 | 2.5 | 0.01 | - | - | - | - | - | - |
| LIC-Biv | 67.1 | 41.45 | GT/NA | -1.7 | 0.04 | 1.8 | 0.01 | - | - | - | - | - | - |
| LIC-Ento | 67.1 | 41.45 | GT/NA | -2.3 | 0.01 | 1.4 | 0.04 | - | - | - | - | - | - |
| VAL-Ager | 67.0 | 42.5 | GT/NA | -1.7 | 0.1 | 1.7 | 0.05 | 0.707 | 0.015 | 19.2 | 5.1 | -0.5 | 1.1 |
| ENC-Pinna | 66.85 | 44.4 | GT/NA | -0.7 | 0.1 | 2.6 | 0.03 | - | - | - | - | - | - |
| ENC-BivA | 66.85 | 44.4 | FeMn/NA | -1.5 | 0.1 | 1.4 | 0.04 | 0.692 | 0.003 | 24.0 | 0.9 | 0.7 | 0.2 |
| ENC-BivB | 66.85 | 44.4 | FeMn/BAD | -1.8 | 0.1 | 1.0 | 0.1 | 0.664 | 0.005 | 34.1 | 2.0 | 2.4 | 0.3 |
| GRO-Biv | 66.7 | 46 | Mn/NA | -1.6 | 0.1 | 1.7 | 0.04 | 0.701 | 0.011 | 20.9 | 3.6 | 0 | 0.8 |
| GRO-Ost | 66.7 | 46 | GT/NA | -1.4 | 0.04 | 2.5 | 0.05 | 0.698 | 0.001 | 22.0 | 0.3 | 0.4 | 0.1 |
| ROT-Acut | 66.55 | 52.7 | GT/NA | -1.4 | 0.2 | 2.6 | 0.1 | 0.710 | 0.005 | 17.8 | 1.8 | -0.5 | 0.5 |
| ROT-Ento | 66.55 | 52.7 | GT/NA | -1.0 | 0.1 | 1.7 | 0.03 | 0.695 | 0.011 | 22.9 | 3.9 | 1.0 | 0.8 |
| LAV-AcutG | 66.4 | 56.75 | GT/GOOD | -1.4 | 0.1 | 3.2 | 0.03 | - | - | - | - | - | - |
| LAV-NeitA | 66.4 | 56.75 | Fe/NA | -1.3 | 0.1 | 1.1 | 0.03 | - | - | - | - | - | - |
| LAV-NeitB | 66.4 | 56.75 | GT/GOOD | -1.3 | 0.02 | 1.6 | 0.01 | - | - | - | - | - | - |
| LAV-AcutA | 66.4 | 56.75 | GT/NA | -0.6 | 0.2 | 2.8 | 0.1 | 0.704 | 0.019 | 20.8 | 6.4 | 0.8 | 1.4 |
| LAV-AcutB | 66.4 | 56.75 | GT/GOOD | -0.8 | 0.1 | 2.8 | 0.2 | 0.699 | 0.022 | 22.3 | 6.9 | 0.9 | 1.5 |
| LAV-AcutC | 66.4 | 56.75 | GT/NA | -1.2 | 0.1 | 3.2 | 0.02 | - | - | - | - | - | - |
| LAV-AcutD | 66.4 | 56.75 | GT/NA | -1.2 | 0.04 | 2.9 | 0.02 | - | - | - | - | - | - |
| LAV-AcutE | 66.4 | 56.75 | GT/NA | 0.0 | 0.05 | 3.3 | 0.03 | 0.738 | 0.013 | 9.5 | 3.9 | -0.9 | 0.8 |
| LAV-AcutF | 66.4 | 56.75 | GT/NA | -1.0 | 0.04 | 2.4 | 0.03 | - | - | - | - | - | - |
| LAV-Ost | 66.4 | 56.75 | GT/NA | 0.7 | 0.1 | 3.0 | 0.04 | - | - | - | - | - | - |
| LAV-Ager | 66.4 | 56.75 | GT/NA | -0.9 | 0.2 | 2.5 | 0.1 | 0.724 | 0.016 | 14.2 | 4.9 | -0.9 | 1.1 |
| LAV-Psop | 66.4 | 56.75 | GT/NA | -1.7 | 0.1 | 2.9 | 0.1 | - | - | - | - | - | - |
| EMAU-AgerA | 66.375 | 60 | GT/NA | -1.2 | 0.04 | 3.3 | 0.04 | - | - | - | - | - | - |
| EMAU-AgerB | 66.375 | 60 | GT/BAD | -1.4 | 0.1 | 3.3 | 0.1 | - | - | - | - | - | - |
| EMAU-AgerC | 66.375 | 60 | GT/GOOD | -1.2 | 0.3 | 3.1 | 0.2 | 0.704 | 0.007 | 19.8 | 2.1 | 0.2 | 0.5 |
| EMAU-AcutA | 66.375 | 60 | NA/NA | -1.3 | 0.2 | 3.1 | 0.1 | 0.704 | 0.014 | 20.1 | 4.4 | 0.1 | 1.0 |
| EMAU-AcutC | 66.375 | 60 | GT/NA | -1.0 | 0.04 | 2.8 | 0.02 | - | - | - | - | - | - |
| EMAU-AcutD | 66.375 | 60 | GT/NA | -1.4 | 0.1 | 2.9 | 0.01 | - | - | - | - | - | - |
| EMAU-AcutE | 66.375 | 60 | GT/NA | -1.3 | 0.1 | 3.0 | 0.04 | - | - | - | - | - | - |
| EMAU-AcutF | 66.375 | 60 | GT/GOOD | -0.5 | 0.01 | 3.4 | 0.01 | - | - | - | - | - | - |
| EMAU-AgerD | 66.375 | 60 | GT/NA | -1.3 | 0.01 | 3.3 | 0.01 | - | - | - | - | - | - |
| EMAU-AcutG | 66.375 | 60 | GT/GOOD | -1.5 | 0.03 | 3.0 | 0.02 | - | - | - | - | - | - |
| EMAU-AcutH | 66.375 | 60 | GT/NA | -1.3 | 0.03 | 4.0 | 0.01 | - | - | - | - | - | - |
| EMAU-Cer | 66.375 | 60 | GT/NA | -0.7 | 0.01 | 1.9 | 0.01 | - | - | - | - | - | - |
| LAU-Unoy | 66.35 | 61.15 | GT/OK | -1.3 | 0.02 | 3.2 | 0.02 | - | - | - | - | - | - |
| LAU-AcutA | 66.35 | 61.15 | FeMn/NA | -0.8 | 0.1 | 3.2 | 0.04 | - | - | - | - | - | - |
| LAU-AcutB | 66.35 | 61.15 | GT/NA | -1.3 | 0.03 | 3.2 | 0.02 | - | - | - | - | - | - |
| LAU-AcutC | 66.35 | 61.15 | GT/NA | -1.2 | 0.02 | 3.2 | 0.01 | - | - | - | - | - | - |
| LAU-AcutD | 66.35 | 61.15 | GT/NA | -1.0 | 0.1 | 3.3 | 0.01 | - | - | - | - | - | - |
| LAU-AcutE | 66.35 | 61.15 | GT/GOOD | -0.7 | 0.1 | 3.4 | 0.04 | - | - | - | - | - | - |
| LAU-AcutF | 66.35 | 61.15 | GT/NA | -1.1 | 0.3 | 2.8 | 0.2 | 0.717 | 0.014 | 15.8 | 4.3 | -0.7 | 1.1 |
| LAU-AcutG | 66.35 | 61.15 | GT/GOOD | -1.3 | 0.1 | 3.3 | 0.03 | 0.698 | 0.003 | 21.8 | 1.2 | 0.4 | 0.3 |
| NEKB-AcutA | 66.325 | 62 | GT/NA | -0.7 | 0.2 | 3.4 | 0.1 | 0.683 | 0.008 | 27.2 | 3.0 | 2.1 | 0.7 |
| NEKB-AcutBC | 66.325 | 62 | GT/GOOD | -1.2 | 0.1 | 3.3 | 0.1 | - | - | - | - | - | - |
| NEKB-AcutD | 66.325 | 62 | GT/GOOD | -1.7 | 0.01 | 3.1 | 0.01 | 0.690 | 0.007 | 24.7 | 2.2 | 0.7 | 0.5 |
| NEKB-AcutE | 66.325 | 62 | GT/BAD | -0.9 | 0.03 | 2.7 | 0.04 | - | - | - | - | - | - |
| NEKB-AcutF | 66.325 | 62 | Fe/NA | -0.5 | 0.02 | 2.8 | 0.01 | - | - | - | - | - | - |
| NEKB-Ager | 66.325 | 62 | GT/NA | -1.2 | 0.1 | 2.6 | 0.03 | 0.689 | 0.003 | 25.1 | 0.9 | 1.2 | 0.2 |
| NEK-Neit | 66.3 | 64 | GT/BAD | -1.4 | 0.1 | 1.8 | 0.1 | - | - | - | - | - | - |
| NEK-Acut | 66.3 | 64 | GT/BAD | - | - | - | - | - | - | - | - | - | - |
| NEK-NeitB | 66.3 | 64 | GT/NA | -1.7 | 0.2 | 1.6 | 0.03 | 0.697 | 0.013 | 22.3 | 4.3 | 0.1 | 0.8 |
| CBR-Acut | 66.275 | 65 | GT/NA | -2.1 | 0.1 | 2.8 | 0.05 | 0.702 | 0.013 | 20.8 | 4.4 | -0.5 | 1.0 |
| CBR-Biv | 66.275 | 65 | GT/NA | -1.2 | 0.1 | 3.1 | 0.1 | 0.706 | 0.003 | 19.4 | 1.0 | 0 | 0.3 |
| MEER-RasA | 66.07 | 76 | GT/GOOD | -2.0 | 0.02 | 2.3 | 0.05 | 0.698 | 0.001 | 21.9 | 0.4 | -0.2 | 0.1 |
| MEER-RasB | 66.07 | 76 | GT/GOOD | -1.3 | 0.2 | 2.1 | 0.1 | 0.719 | 0.019 | 15.9 | 5.6 | -0.9 | 1.2 |
| MEER-mat | 66.07 | 76 | GT/NA | -0.1 | 0.1 | 1.3 | 0.04 | 0.705 | 0.011 | 19.8 | 3.5 | 1.2 | 0.8 |
| GCAV-OystA | 66.0 | 85 | GT/GOOD | -2.3 | 0.1 | 2.7 | 0.02 | 0.699 | 0.003 | 21.6 | 1.0 | -0.6 | 0.2 |

The remaining 27 samples plus seven of the above 39 were analyzed for $\delta^{18}O$ and $\delta^{13}C$ in conjunction with clumped isotope analysis in the University of Michigan Stable and Clumped Isotopes for Paleoclimatology and Paleoceanography (SCIPP) Laboratory (Fig. S1, Fig. S2, Fig, S3, Table 1). During clumped isotope analysis, one measurement simultaneously acquires

$\Delta_{47}$, which is a function of carbonate formation temperature (or past seawater temperature in the case of fossil shells), $\delta^{13}C$, and $\delta^{18}O$ of $CO_2$ gas evolved from $CaCO_3$ (Eiler, 2011). Samples are typically replicated a minimum of three times, and replicates are averaged to calculate a sample mean. Here, after combining all replicates for each sample, individual replicates were deemed to be outliers if they fell more than four standard deviations outside of the mean of the remaining three to four replicates for a given sample. A total of 108 replicates representing 34 unique samples were analyzed over a span of seven

years using two different machines and procedures (see below). Only samples with at least three acceptable replicates were considered (n = 27 shells + one matrix sample prior to preservation screening) (Table 1).

Between 2016 and 2020, a first set of fossil shell powders were measured for clumped isotopic composition ($\Delta_{47}$) in the University of Michigan Stable Isotope Lab using a Thermo-Finnigan MAT 253 dual inlet isotope ratio mass spectrometer and

a manual sample preparation device described in detail by Defliese et al. (2015), with updates to Porapak temperature outlined by Petersen et al. (2016c). Briefly, 3.5–5 mg of powder is reacted at 75 °C in a common acid bath, and evolved $CO_2$ is cryogenically separated from water and other contaminants through two -80 °C ethanol-liquid nitrogen traps and one PorapakQ/silver wool trap. Clean $CO_2$ is measured for five acquisitions of 12 cycles at a major beam (m/z 44) voltage of 16 V. Raw voltages of masses 44 to 49 were converted to bulk isotopic values ($\delta^{45}$ to $\delta^{49}$) and clumped isotopic values ($\Delta_{47}$, $\Delta_{48}$,

and $\Delta_{49}$) using IUPAC/Brand parameters using an R code presented in Petersen et al. (2019). Raw $\Delta_{47}$ values were placed in a gas-based absolute reference frame following methods introduced by Dennis et al. (2011), using theoretical equilibrium values for heated (1000 °C) and water-equilibrated (25 °C) standard gases (0.0266 and 0.9198, respectively) and the 75 °C acid fractionation factor of +0.072 ‰ for $\Delta_{47}$ from Petersen et al. (2019). Boundaries between each one-to-three-week-long correction window were selected to account for drift in equilibrium gas line slopes and to maintain consistent values of in-

house carbonate standards through time. $\delta^{18}O$ of carbonate ($\delta^{18}O_{carb}$) was calculated from $\delta^{18}O$ of evolved $CO_2$ using an empirical acid fractionation factor defined using in-house carbonate standards independently calibrated to NBS-18 and NBS-19 (alpha = 1.007888 for calcite at 75 °C) (Table S4). Long-term standard deviation of $\delta^{13}C$, $\delta^{18}O$, and $\Delta_{47}$ on the MAT253 were consistent across time at approximately 0.1 ‰, 0.3 ‰, and 0.020 ‰, respectively, based on in-house carbonate standards run many times over the measurement interval (Table S4).


In 2021 and 2022, additional powders were analyzed in the University of Michigan SCIPP lab using a Nu Perspective isotope dual inlet ratio mass spectrometer connected to a NuCarb automated sample preparation device. This device reacts sample powders in individual vials under vacuum at 70 °C by slowly injecting 150 uL of 103 % phosphoric acid. Vials are isolated for the first five minutes of the reaction to reduce excess bubbling, then opened to a variable temperature trap held at -160 °C

for 15 additional minutes, continuously collecting evolved $CO_2$. Next, this variable-temperature trap is elevated to -60 °C,

releasing $CO_2$ to pass through a static trap filled with Porapak Q material held at -30 °C. $CO_2$ is frozen on the far size in another variable temperature trap set at -160 °C for 800 seconds. $CO_2$ is warmed up to the gas phase, and the yield is recorded with a transducer. The Nu Perspective and NuCarb can handle both larger (3–6 mg) and smaller (350–450 ug) samples. Following purification, depending on the transducer reading, the automated sequence directs $CO_2$ from a larger sample to equilibrate into

the bellows or freezes $CO_2$ from a smaller sample into a cold finger between the bellows and the change-over block. All sample replicates in this study were between 3–5 mg in size and were therefore analysed in "bellows mode". Previous studies using a Nu Perspective and NuCarb for clumped isotope analysis described "cold-finger mode" (Mackey et al., 2020). In bellows mode, the bellows are compressed until a desired beam strength of 80 nA is achieved on the major (m/z 44) ion beam. Gas is analyzed for four blocks of 20 reference-sample cycles, with 20 seconds of integration on each half cycle. Bellows are

continually adjusted between each cycle to maintain the initial beam strength at all times.

As above, raw beam intensities of masses 44–49 were converted to bulk isotopic values ($\delta^{45}$ to $\delta^{49}$) and clumped isotopic values ($\Delta_{47}$, $\Delta_{48}$, and $\Delta_{49}$) using IUPAC/Brand parameters using an R code presented in Petersen et al. (2019), adjusted slightly to accommodate differing data formats from the Nu instrument. Isotopic values were converted into the Intercarb Carbon Dioxide

Equilibrium Scale (I-CDES25) absolute reference frame using $\Delta_{47}$ values for four ETH standards defined by the Intercarb project (Meckler et al., 2014; Bernasconi et al., 2021) and a $\Delta_{47}$ acid fractionation factor of +0.066 ‰ for 70 °C from Petersen et al. (2019). First, a single slope was fitted through ETH 1 and ETH 2 (adjusted by 0.0033 ‰) in $\delta^{47}$ vs. $\Delta_{47}$ space, then all four ETH standards were used for the empirical transfer function step. Some heated gases (1000 °C and 200 °C) and equilibrated gases (25 °C) were analyzed for cross comparison but were not used in reference frame calculations. Carbonate

$\delta^{18}O$ and $\delta^{13}C$ were each calculated using a single-step transfer function defined by the four ETH standards, using values published in Bernasconi et al. (2018). Although use of ETH standards does not require an acid fractionation factor when applied to an entire dataset, to combine data from gas-based and carbonate-based reference frames requires normalization to the same acid digestion temperature. Long-term standard deviation of $\delta^{13}C$, $\delta^{18}O$, and $\Delta_{47}$ on this instrument were consistent across time at better than 0.05 ‰, 0.1 ‰, and 0.018 ‰, respectively (Table S4).


Clumped isotope ($\Delta_{47}$) values for between two and five in-house carbonate standards were tracked to ensure consistency and comparability of isotope values through time and across machines. Stable isotope ($\delta^{13}C$ and $\delta^{18}O$) values for in-house carbonate standards were determined relative to NBS-18 and NBS-19 on the same Kiel IV + MAT253 used for bulk sample $\delta^{13}C$ and $\delta^{18}O$ (Table S4). $\Delta_{47}$ values for these standards were defined based on comparison to gas standards over multiple years (2015

and 2018–2019 for standards CM, OO, and CORS) or comparison to ETH standards over a six-month measurement session in 2020 (for all five) (Table S4). Close agreement in $\Delta_{47}$, $\delta^{18}O$, and $\delta^{13}C$ between long-term values and session specific values on all three instruments confirms that data from different measurement sessions and instruments can be directly compared (Table S4). Comparison of replicates of the same sample powder run on each machine support this.

Finally, acid-corrected $\Delta_{47}$ values (CDES25) from both measurement periods were converted to temperature using the composite calibration of synthetic carbonates from Petersen et al. (2019) [$Temp(°C) = (0.0383*10^6)/(\Delta_{47}–0.258)–273.15$]. This equation was chosen in favor of the ETH-standard-based calibration of Anderson et al. (2021) due to the use of a gas-based reference frame for the early measurements and the fact that calibration data acquired on the MAT 253 by Defliese et al. (2015) and Winkelstern et al. (2016) are included in the Petersen et al. (2019) composite calibration equation. Use of the Anderson et

al. (2021) equation results in temperatures 1.5–2.5 °C cooler and $\delta^{18}O_{sw}$ values proportionally lower, but doesn't change any of the conclusions in this study. Seawater $\delta^{18}O$ ($\delta^{18}O_{sw}$) was calculated for shell specimens using measured carbonate $\delta^{18}O$ and $\Delta_{47}$-derived temperatures and the calcite-water relationship described by Kim and O'Neil (1997). All $\delta^{18}O_{sw}$ values are reported on the VSMOW scale (Table 1).

Long-term reproducibility (1SD) of $\Delta_{47}$ in carbonate standards (0.020 ‰ for MAT253 and 0.018 ‰ for Nu) was used to calculate a long-term standard error in $\Delta_{47}$ for each sample. For temperature and $\delta^{18}O_{sw}$ values, a long-term standard error was calculated using the long-term standard error in $\Delta_{47}$ and the pseudo-linear relationship between internal 1SE on $\Delta_{47}$ and either temperature or $\delta^{18}O_{sw}$ in all samples [for this study: long-term 1SE in temperature = 306.7 * (long-term 1SE in $\Delta_{47}$) + 0.1778, and long-term 1SE in $\delta^{18}O_{sw}$ = 66.33 * (long-term 1SE in $\Delta_{47}$) + 0.0461]. The external 1SE for all three parameters was selected

as the larger of the internal and long-term standard error. All replicate-level clumped isotope and stable isotope data has been submitted to the permanent data archive EarthChem as part of the ClumpDB database.

## 4 Results

### 4.1 Preservation and sample screening

Isotopic signatures may be affected by both bulk chemical alteration (dissolution and recrystallization replacing original shell material and/or overgrowth of secondary material changing the bulk isotopic composition) and solid-state bond reordering (changes in the number of heavy isotope clumps in the crystal lattice and overprints the original recorded temperature without any mass transfer). Overall, SEM preservation ratings and trace element concentrations were low in most samples (Fig. 3, Fig. 4, Table S3), ruling out significant chemical alteration, and a mild thermal history suggests reordering is not a concern.


    In the 21 shells imaged for SEM, 6 displayed evidence of secondary growth (SG = 2 or 3) and were not considered further. One sample showed evidence of dissolution but no secondary growth and was deemed "okay" and included in interpretations. 14 samples showed little to no evidence of secondary growth (SG = 0 or 1) and were deemed "good" and included. In the 65 shell specimens tested for trace elements, Mn concentration ranges from 17–210 ppm, while Fe concentration ranges from 96–

7151 ppm. The matrix sample was measured to have 60 ppm Mn and 664 ppm Fe. Threshold values of 100 ppm and 2050 ppm were selected for Mn and Fe, respectively, to exclude most samples that failed SEM screening, while including all samples

that exhibited only original material under SEM (Fig. 3, Fig. 4). These thresholds are more conservative than those proposed by other studies (Morrison and Brand, 1987; Voigt et al., 2003; Ullmann et al., 2013, de Winter et al., 2018), but, in total, still only result in the flagging of nine specimens out of the total of 65 (Fig. 4, Table S3).


Bond reordering without exchange of mass can occur if samples are heated above 100–150 °C (Henkes et al., 2014; Winkelstern and Lohmann, 2016), such as during burial. Heating breaks and reforms bonds between atoms in the carbonate lattice and changes the clumped isotopic composition—and therefore interpreted temperature—without affecting the bulk $\delta^{18}O$ or $\delta^{13}C$ composition appreciably. Regional geologic evidence suggests a relatively shallow burial history for these strata. In the southern Netherlands, the Roer Valley Graben contains several basin-wide unconformities dated to the Late Cretaceous and middle Paleocene that signify two basin inversions, leaving the Maastrichtian and Danian strata relatively undisturbed (Luijendijk et al., 2011 and references therein). Outstandingly well-preserved mosasaur skeletons and hollow aragonitic molds in the Maastricht Formation further indicate a lack of burial or significant diagenetic fluid flow during this time (Dortangs et al., 2002; Jagt et al., 2016). Overall, these burial conditions are not expected to result in bond reordering in any samples.


Isotopic evidence supports that selected trace element and SEM thresholds are conservative and that overall diagenetic alteration in this region is mild to absent. Despite being excluded based on trace element and/or SEM results, the isotopic values of excluded samples were often not substantially different than other shells that passed diagenesis screening (e.g., only one of four measured for clumped isotopes had a higher-than-normal $\Delta_{47}$-derived temperature) (Fig. 5). Additionally, the matrix sample taken from the Meerssen Member showed trace element concentrations below thresholds (Fig. 4) and a temperature of 19.8 ± 3.7 °C (Fig. 5), very similar to temperatures from fossil shells, indicating that the bulk matrix—which is more likely to be affected by recrystallization or reordering than a dense calcite shell—has also not undergone noticeable isotopic or geochemical alteration. This also explains how three samples failed SEM screening (Fig. 3) but showed low trace element concentrations (Fig. 4). We infer that in these samples, although some secondary growth was seen under SEM (Fig. 3), it must have been volumetrically insignificant in the total powdered sample, not present in the portion of the shell that was powdered (which differed from the portion imaged under SEM), and/or did not have elevated trace element concentrations.

In total, 12 of 65 shell samples failed one or both screening technique, with nine failing trace element thresholds, six failing SEM criteria, with three of the six or nine failing both. The one sample that was not measured was assumed to pass screening, based on the majority of samples passing screening. An additional three samples failed stable isotope measurements for analytical reasons, and two more failed to achieve the three good replicates necessary for $\Delta_{47}$ but were still used for $\delta^{18}O$ and $\delta^{13}C$. In the six samples that were measured on both the Kiel and one of the clumped isotope machines, clumped isotope derived $\delta^{18}O$ and $\delta^{13}C$ values were used (but only differed by 0.1–0.2 ‰ from Kiel-derived values). After removing poorly-preserved and insufficiently analyzed samples, the dataset includes 51 well-preserved shell samples analyzed for $\delta^{18}O$ and

$\delta^{13}C$, and 23 well-preserved shell samples with acceptable $\Delta_{47}$ analyses (Table 1). All further interpretation excludes discussion of removed samples.

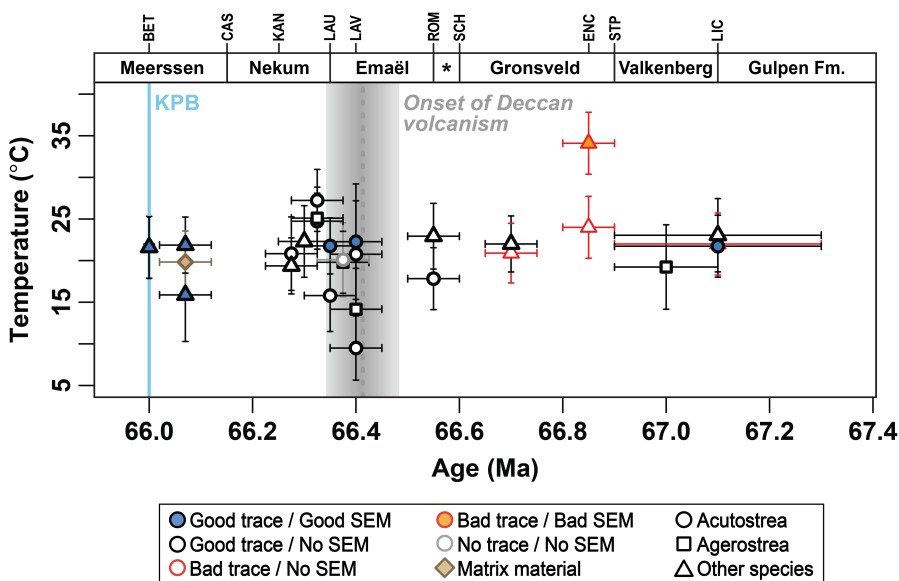

**Figure 5: Maastrichtian $\Delta_{47}$-derived paleotemperatures for all samples.** Samples shown in red (n=4) did not pass screening thresholds and were not included in further interpretation. Error bars represent the larger of either the internal or long-term
1SE (see text). Onset of the main Deccan Traps eruptions (grey shading) at 66.413 ± 0.067 Ma (Sprain et al., 2019). Member names and horizon abbreviations are labeled across top of figure (see Fig. 2). *Schiepersberg Member

**4.2 Stable isotopes**

The 51 shell specimens exhibit a range of carbonate $\delta^{18}O$ ($\delta^{18}O_{carb}$) values between -2.3 ‰ and -0.5 ‰ VPDB, with two
samples showing higher values of 0.0 and 0.7 ‰ VPDB (Fig. S1). The average $\delta^{18}O_{carb}$ value throughout the section is -1.2 ± 0.5 ‰ (1SD) VPDB. In contrast, the carbonate fraction of bulk matrix in the one sample from the Meerssen Member measured here and in a high-resolution record from the Lichtenberg Horizon to the top of the Nekum Member (Vellekoop et al., 2022) fall anywhere from ~0 to ~1 ‰ higher than molluscan carbonate (Fig. S1), with a mean value of -0.6 ± 0.4 ‰ (1SD) VPDB and a range of -1.7 ‰ to 0.0 ‰ VPDB through the section.


Carbonate $\delta^{13}C$ ($\delta^{13}C_{carb}$) values range from 1.4 ‰ to 3.4 ‰ VPDB, with one high sample at 4.0 ‰ VPDB (Fig. S2). The average $\delta^{13}C_{carb}$ value is 2.8 ± 0.6 ‰ (1SD) VPDB. Bulk matrix $\delta^{13}C$ is clearly lower than most molluscan carbonate, by anywhere from 1 to 2 ‰ VPDB (Fig. S2). The average bulk matrix $\delta^{13}C$ value is 1.7 ± 0.1 ‰ (1SD) VPDB with a range of 1.3 ‰ to 1.9 ‰ VPDB through the section.


In most cases, the number of specimens representing each taxon is too small and/or the stratigraphic range is too limited to make reliable inter-species comparisons (e.g., *Rastellum* sp.). Even considering this, the pectinids *Entolium membranaceum* and *Neithea regularis* appear to record carbon isotope compositions ~1–1.5 ‰ more depleted relative to other species, which mainly represent Ostreida, while oxygen isotopic compositions appear similar (Fig. S3). This inter-order difference may represent differing use of metabolic (respired) carbon vs. environmental carbon between these two orders of bivalvia. Between the two most common taxa (*Acutostrea uncinella* and *Agerostrea ungulate*—both oysters), there does not appear to be any systematic offset in oxygen or carbon isotopic compositions (Fig. S3).

## 4.3 Temperature estimates

Four of the 27 shell specimens successfully measured for $\Delta_{47}$ were excluded from further interpretation due to evidence of diagenesis (described above). The 23 samples that pass diagenetic screening criteria exhibit paleotemperatures ranging from 9.5–27.2 °C, with a mean value of 20.4 ± 3.8 °C (1SD) (Fig. 5). In all instances where multiple taxa were sampled from the same horizon, measured temperatures are within error of each other, suggesting no inter-species offset due to vital effects. In the absence of vital effects in any taxa, all species are expected to record the same temperature due to their common benthic habitat. The high agreement between taxa from the same horizon seen here is similar to that seen in other deposits of the same age from the Gulf Coastal Plain (Meyer et al., 2018), although further replication to reduce error bar size would be necessary to rule out small vital effects.

In most horizons with multiple samples, estimated temperatures overlap within error for all samples (Fig. 5). The exception is the Lava Horizon (66.4 Ma), which shows a large spread in temperatures from 9.5 ± 3.8 °C (external 1SE) to 22.3 ± 6.9 °C (external 1SE) (Fig. 5). This may represent seasonal aliasing in samples (e.g., powdering a portion of shell representing wintertime growth vs. summertime growth), although these samples were not treated any differently than other horizons where this is not observed. This may also represent long-term time averaging into a single horizon, where individual shells may have lived thousands of years apart. It is not a species-related offset, as the highest and lowest temperatures both occur in *Acutostrea uncinella* (Fig. 5). A similar argument can be extended to other horizons where a larger spread is seen, such as the lower Meersen Member (66.1 Ma) and the Romontbos Horizon (66.55 Ma), although samples in these layers are still within error at the external 1SE level.

A fluctuating temporal trend is observed in which paleotemperatures rise from a low at 66.4 Ma to a peak in the basal Nekum Member at 66.325 Ma, followed by a decrease by 66.275 Ma that persists through the KPB (Fig. 5). The onset of warming coincides with the onset of Deccan volcanism 66.413 ± 0.067 Ma (Sprain et al., 2019) (Fig. 5), within the error of our age model (Table S1). We therefore identify this warming event as the named "Late Maastrichtian Warming Event" (LMWE) (Woelders et al., 2018, Hull et al., 2020).

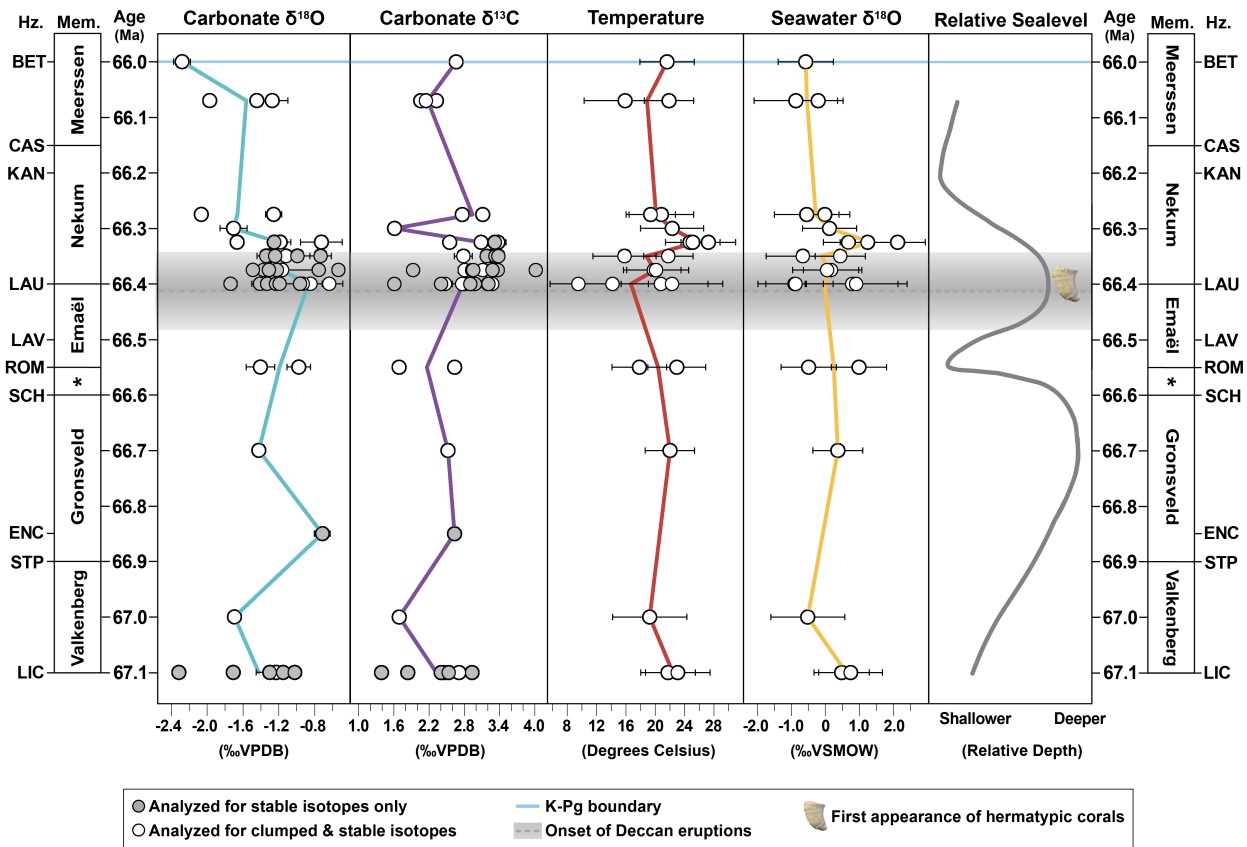

**Figure 6: Isotopic compositions and paleotemperatures of fossil shells compared to inferred relative sea level in the Maastricht Formation.** Error bars represent 1SD for $\delta^{18}O$ (a) and $\delta^{13}C$ (b), and external 1SE for temperature (c) and $\delta^{18}O_{sw}$ (d) (see section 3.3). All samples failing diagenetic screening criteria have been removed. Colored lines pass through horizon means. Age model from Vellekoop et al. (2022.) (Table S3). The relative sea level curve (e) is based on Schiøler et al. (1997), adjusted to the age model used in this study. Hermatypic coral appearance data (e) is from Leloux (1999). Onset of the main Deccan Traps eruptions (grey dotted line) at $66.413 \pm 0.067$ Ma (Sprain et al., 2019).

## 4.4 Seawater $\delta^{18}O$ estimates

Calculated $\delta^{18}O_{sw}$ values ranged from -0.9 to +2.1 ‰ VSMOW (Fig. S4). The average $\delta^{18}O_{sw}$ value is +0.2 ± 0.8 ‰ (1SD) VSMOW, above the ice-free mean ocean water estimate of -1.0 ‰ VSMOW. This heavier average $\delta^{18}O_{sw}$ value is consistent with the presence of some continental ice volume (presumed to be located on the Antarctic continent) during the "cool greenhouse" Maastrichtian interval (Miller et al., 2005; Petersen et al., 2016a; Ladant and Donnadieu, 2016). Just like with temperature, where multiple taxa were sampled within a given horizon, $\delta^{18}O_{sw}$ values are within error of each other, although there is fairly large spread in the Lava Horizon again (Fig. S4).

Sea water $\delta^{18}O$ values show a similar temporal trend to that of temperature (Fig. 6c, d). We observe peak $\delta^{18}O_{sw}$ values at the base of the Nekum Member (1.4 ± 0.7 ‰ VSMOW, horizon mean and 1SD of three samples) and the lowest $\delta^{18}O_{sw}$ values occur at the Lava Horizon in the Emaël Member (-1.0 ± 1.8 ‰ VSMOW, horizon mean and 1SD of three samples) (Fig. 6d, Fig. 4S). This relationship is not a function of calculations of temperature or $\delta^{18}O_{sw}$ and, instead, falls naturally out of the $\delta^{18}O_{carb}$-$\delta^{18}O_{sw}$-temperature relationship. Across the entire dataset, both temperature and $\delta^{18}O_{sw}$ show weak correlation with $\delta^{18}O_{carb}$ with neither as the primary, sole driver (Fig. S5).

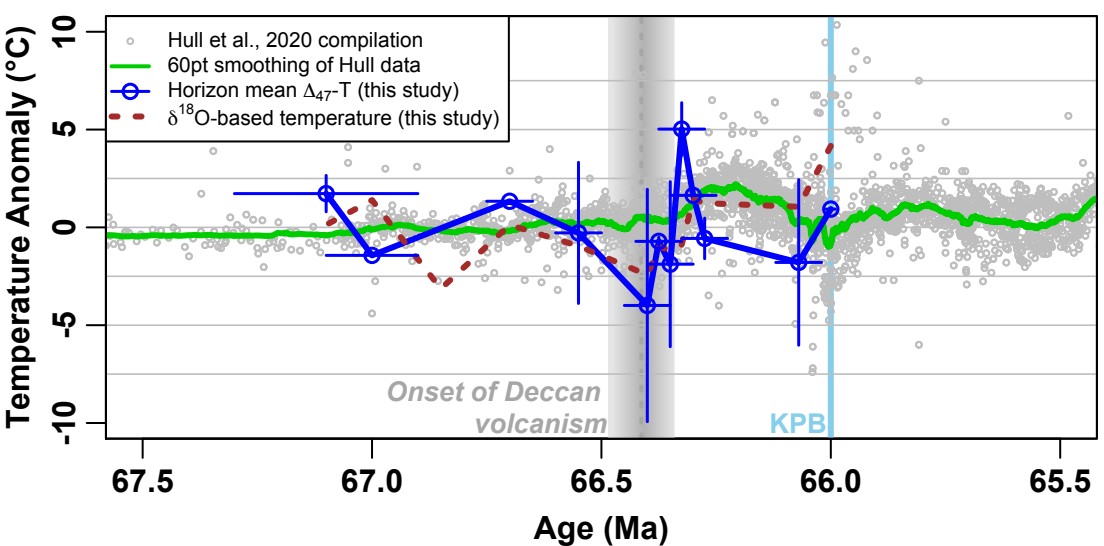

**Figure 7: Temperature evolution through the Late Maastrichtian Warming Event**. Comparison of new temperature data to the compilation of late Maastrichtian temperatures by Hull et al. (2020). Grey and light blue vertical bars indicate onset of Deccan volcanism and the KPB, as in Fig. 5. Vertical error bars on horizon averages represents 1SD of all temperatures from a given horizon. Horizons with no error had only one sample. Horizontal error bars represent uncertainty in the age model of Vellekoop et al. (2022). Brown dashed line indicates a temperature profile calculated from $\delta^{18}O_{carb}$ assuming a fixed $\delta^{18}O_{sw}$ value of -1 ‰ VSMOW, and, similarly to the Hull et al. (2020) composite record, shows peak temperatures delayed relative to the $\Delta_{47}$-based temperature peak.

## 5 Discussion

### 5.1 Further evidence for a warmer Maastrichtian world and a Late Maastrichtian Warming Event

New data presented here indicate that mean annual ocean temperatures in the Maastricht region (paleolatitude ~40° N; van Hinsbergen et al., 2015) varied from 9.5 ± 3.9 °C to 27.2 ± 3.7 °C between 67.1 and 66.0 Ma, with a mean temperature of 20.4 ± 3.8 °C (1SD) over this interval. This is much warmer than the modern-day climate of Maastricht, which has a mean annual air temperature of 9.8 °C (Maastricht Climate, 2021). Warmer-than-modern temperatures in the Maastrichtian have been seen before in many studies (Table S1) and are expected due to elevated atmospheric $CO_2$ levels during the Maastrichtian of ~1000

ppm on average over the study interval (Zhang et al., 2018; Henahan et al., 2019; Hoenisch, 2021). Reconstructed Maastrichtian marine temperatures from sites located between 30–50° N show an average temperature of 19.8 ± 5.2 °C (1SD, n = 23 studies; Table S1), in close agreement with our site mean. Outside the marine realm, Maastrichtian-aged fossil plants also suggest that terrestrial temperatures at 40° N were warmer, measuring around 15 °C (Golovneva, 2000), and that near 49° N, air temperatures varied from 10–18 °C during this time interval (Wilf et al., 2003). General circulation models using Late Cretaceous paleogeography and elevated atmospheric $pCO_2$ levels also display similarly warmer climate globally as a result of $CO_2$-induced warming (Tabor et al., 2016; Ladant et al., 2020). Modeled zonal mean temperatures at a paleolatitude of 40° N are 20.2 °C for a 2xCO$_2$ (560 ppm) Maastrichtian scenario or 21.4 °C for a 4xCO$_2$ (1120 ppm) Maastrichtian scenario (Tabor et al., 2016).

Multiple paleoclimate records document a Late Maastrichtian Warming Event (LMWE) (e.g., Stott and Kennett, 1990; Li and Keller, 1998; Petersen et al., 2016a; Woelders et al., 2017; Woelders et al., 2018; Barnet et al., 2019; Gao et al., 2021; Nava et al., 2021), beginning gradually at 66.4 Ma with more rapid warming at 66.3 Ma (Hull et al., 2020) (Fig. 7). The timing of this LMWE appears to closely follow the onset of LIP volcanism in the Indian subcontinent, based on $^{40}$Ar-$^{39}$Ar ages for the oldest basalts of the Deccan Traps (66.413 ± 0.067 Ma from the Jawhar Formation; Sprain et al., 2019) and shifts in marine osmium isotope ratios towards mantle values beginning ~400 kyr prior to the KPB (Ravizza and Peucker-Ehrenbrink, 2003; Hull et al., 2020), although others argue for a slightly later onset around 66.311 ± 0.051 Ma (Schoene et al., 2019; Nava et al., 2021). Volcanically-derived $CO_2$ is widely considered to be the cause of this warming.

The $\Delta_{47}$-derived temperature record generated here documents a warming in the Nekum Member, beginning from a temperature low in the Lava Horizon at 66.4 ± 0.05 Ma (16.6 ± 5.9 °C, 1SD, N=4) and increasing to the basal Nekum at 66.325 ± 0.05 Ma (25.7 ± 1.4 °C, 1SD, N=3) over 75 kyrs (Fig. 5). The magnitude of this warming is calculated as 9.1 °C going from these listed horizons or, more conservatively, 5.8 °C comparing the warmest horizon (basal Nekum) to a pre-event baseline averaging the preceding seven horizons (19.9 ± 2.0 °C, N=14). Although this warming event is only barely statistically distinguishable as differing from the mean of the full record when considering the size and meaning of the 1SE error bars and the fairly small number of samples representing peak warmth (n=3) (Fig. 5), the close agreement between all three points from this horizon and the following additional lines of evidence support the interpretation of this warming as an expression of the LMWE.

Based on a newly updated age model (Vellekoop et al., 2022) (Fig. S1), the warmest $\Delta_{47}$-derived temperatures slightly follow or align with the onset of earliest Deccan eruptions (depending on whether a date of 66.413 Ma or 66.3 Ma is used) and the general age of the globally defined LMWE (Hull et al., 2020) (Fig. 7). The dinocyst *Palynodinium grallator*—a marker for the LMWE at northern mid-latitudes (Vellekoop et al., 2018; 2019)—first appears in the Maastricht Formation in the Nekum

Member (Schioler et al., 1997; Vellekoop et al., 2019). Thermophilic hermatypic scleratinian macrofossils also first emerge in high abundances in the Nekum Member (Liebau, 1978; Leloux, 1999) (Fig. 6e). Together, these records suggest that the LMWE is preserved in the Nekum Member at ENCI quarry and that the warmer $\Delta_{47}$-derived temperatures found there are likely real.

Interestingly, a $\delta^{18}O_{carb}$-based temperature reconstruction assuming constant $\delta^{18}O_{sw}$ (red dashed line, Fig. 7) incorrectly shows a slightly later peak in temperature. The true peak, as indicated by the $\Delta_{47}$-derived record, is hidden in $\delta^{18}O_{carb}$ due to simultaneous increases in temperature and enrichment in $\delta^{18}O_{sw}$ (Fig. 5, Fig. 6, Fig. S4). The timing of warming in our $\Delta_{47}$-derived record appears to predate that of the composite record of Hull et al. (2020) as well (Fig. 7), although uncertainty in age in both records may not sustain this relationship. If age models hold as shown, the later timing seen in the composite record may reflect the same issue of a hidden LMWE peak, as many of the records included are based on foraminiferal $\delta^{18}O_{carb}$ and include an assumption of constant $\delta^{18}O_{sw}$. Another $\Delta_{47}$-derived temperature record across this period from Seymour Island, Antarctica (Petersen et al., 2016a) also shows simultaneous increases in temperature and $\delta^{18}O_{sw}$ and a delayed apparent warming in a $\delta^{18}O_{carb}$-based temperature reconstruction, although the absolute timing of the LMWE in that record is apparently later (~66.24–66.2 Ma). Faulty assumptions of constant sedimentation rate between magnetostratigraphic datums could potentially explain the difference in absolute timing.

Following on the assumed identification of the LMWE in the basal Nekum Member, our temperature record indicates that seawater temperatures in this region returned to baseline levels by the middle Nekum Member, within ~100kyr (Fig. 5). This is consistent with the timing of silicate weathering feedbacks, as previously suggested (Caldiera and Rampino, 1990; Dessert et al., 2001; Petersen et al., 2016a; Tobin et al., 2017; Hull et al., 2020), further supporting the volcanically-derived $CO_2$ as the source of the initial warming event. Others have suggested that the emplacement of large igneous provinces can actually be a long-term $CO_2$ sink and lead to longer-term global climate cooling (Schaller et al., 2012; Johansson et al., 2018). This is possibly visible in the slight cooling between the middle Nekum and Meerssen Members (Fig. 5), although the uncertainty in the Meerssen Member is large.

## 5.2 Ocean circulation, sea level, and/or precipitation changes

Paleogeographic reconstructions (Engelke et al., 2017; Scotese, 2021) (Fig. 1a) and presence of seagrass fossils (Hart et al., 2016; van der Ham et al., 2017) indicate that during the Campanian-Maastrichtian, the study site was located on a large, shallowly submerged carbonate platform. Over the Late Cretaceous, eustatic sea level fluctuated multiple times, including one sea level transgression (KMa5) in the latest Maastrichtian at 66.8 Ma (Haq, 2014). Sea level based on the relative abundance of the palynomorph *Paralecaniella* in ENCI quarry sediments—the same section as our fossils—indicate local sea level fluctuations that differ from the global eustatic curve (Schiøler et al., 1997) (Fig. 6e). Since high relative abundances of this acritarch taxon are suggested to reflect marginal-marine to restricted-marine conditions under high hydrodynamic conditions

(Brinkhuis and Schiøler 1996; Schiøler et al., 1997), observed fluctuations are interpreted as local sea level changes. Alternatively, environmental factors other than sea level might have driven the observed high abundances of *Paralecaniella* in the type-Maastrichtian, such as biological responses to local changes in hydrodynamic conditions and salinity, which would also be reflected in the $\delta^{18}O_{sw}$ record presented here. Nevertheless, seagrass and foraminifera indicate that water depths remained less than 20 m at all times, and often shallower (Hart et al., 2016).

Although this local sea level record derived from palynological abundances may have some issues at a fine scale related to minimal consideration of sedimentology and geochemistry (Vellekoop et al., 2022), we see a notable correlation between it and many of our isotopic parameters (Fig. 6). Peak $\delta^{18}O_{sw}$ values, high temperatures, and, to a lesser degree, higher $\delta^{13}C_{carb}$ values correlate to periods of maximum sea level, while lower $\delta^{18}O_{sw}$ values, cooler temperatures, and lower $\delta^{13}C_{carb}$ values correspond to periods of relatively lower sea level (Fig. 6). In particular, sea level is interpreted to rise from the Lava Horizon up to the Laumont Horizon, peaking around the base of the Nekum Member and then declining again by the Kanne Horizon (Fig. 6e). This aligns with the temperature and $\delta^{18}O_{sw}$ peak in the basal Nekum highlighted above (Fig. 6c, d), coinciding with the onset of Deccan volcanism. A second interval of peak sea level occurs during the Gronsveld Member (Fig. 6e), which also displays temperatures and $\delta^{18}O_{sw}$ values slightly higher than preceding and following horizons (Fig. 6c, d). $\delta^{13}C_{carb}$ values also rise into the Gronsveld Member, and again rise from the Lava to Laumont Horizons (Fig. 6b).

We propose that the observed correlation between environmental conditions and reconstructed local sea level is the result of fluctuating ocean circulation patterns. Paleogeographic reconstructions suggest the presence of a seaway connecting the Arctic Ocean to this study site during the late Maastrichtian (Engelke et al., 2017; Scotese, 2021) (Fig. 1a). Modeling studies show that increased isolation of the Arctic made it fresher during the Maastrichtian (Ladant et al., 2020), suggesting that this boreal water mass was both cooler in temperature and depleted in $\delta^{18}O_{sw}$. As a result, the Maastrichtian type region would have been under the influence of relatively cool, depleted water masses. Our study site was situated on a shallow carbonate platform surrounded by small landmasses, the exact size and position of which are difficult to define (Engelke et al., 2017; Scotese, 2021) (Fig. 1a). During periods of low relative sea level, the land masses of the Rhenohercynian Zone (Rhenish Massif, London-Brabant Massif) would have blocked southern currents originating in warmer regions, which would likely be saltier and have higher $\delta^{18}O_{sw}$ values due to evaporative enrichment (e.g. the semi-restricted Western Tethys) (Fig. 1a). However, as large parts of these land masses were relatively flat and low-lying during the Upper Cretaceous (Vandenberghe et al., 2014), relatively minor increases in sea level could potentially breach sills or island barriers, changing both the position of ocean currents and the dominant water mass in a given area. Indeed, lithostratigraphic studies in the region have shown that during various phases in the Campanian-Maastrichtian, the structural highs of the nearby London-Brabant Massif were fully submerged (e.g., Dusar and Lagrou, 2007). As a result, during sea level lows, the Maastrichtian type region would have been predominantly under the influence of colder, depleted water masses sourcing from the north, while during sea level highs, the influence of warmer, more enriched water masses sourcing from the south likely increased (Fig. 1a).

Similar interactions between cooler and warmer water masses have been called upon to explain unexpected distributions in warm-water ammonite fauna in the Coniacian-Santonian of Europe (Remin et al., 2016) and remarkably southerly occurrences of cold-water belemnites in late Cenomanian France (Gale and Christensen, 1996). In this case, the authors suggest that warm waters flowing east to west from the proto-Tethys competed with cooler waters flowing southward between Greenland and

Scandinavia (see their Fig. 10). We suggest a similar mechanism applied to late Maastrichtian paleogeography and water masses (Fig. 1a). Ostracod assemblage changes indicate increasing contribution of warm Mediterranean/Tethyan waters in the latest Maastrichtian (Bless, 1988). Faunal abundance differences along the US East Coast indicate the presence of an ancient Gulf Stream-like current as far back in time as the Maastrichtian (Watkins and Self-Trail, 2005), which would have warmed the Maastricht region above its expected temperature for latitude when bathed in water from the south, as the modern-day Gulf

Stream does for western Europe.

Fluctuations in $\delta^{13}C$ appear to correlate with changes in temperature and $\delta^{18}O_{sw}$ (Fig. 6). In this hypothetical model of alternating influence of southern and northern source waters, this could indicate different $\delta^{13}C$ of dissolved inorganic carbon ($\delta^{13}C_{DIC}$) in the two water masses. The exact values of $\delta^{13}C_{DIC}$ in either water mass are difficult to define because of

overprinting by $\delta^{13}C$ vital effects in some taxa (*Neithia* sp.*, Entollium* sp.) (Fig. S2, Fig. S3). Looking at the most common taxa (*Acutostrea*), variations on the order of 0.5 ‰ VPDB—or possibly as much as 1 ‰—are possible between horizons. This is similar to variability seen in the modern Atlantic Ocean (Eide et al., 2017) and is therefore plausibly driven by alternating water masses of differing $\delta^{13}C_{DIC}$. In this scenario, the colder, northern (Arctic) water mass would be more depleted in $\delta^{13}C_{DIC}$ than the southern-source water.

This dataset is not capable of distinguishing between the above, preferred hypothesis and a temporally variable input of isotopically depleted freshwater. During times of lower sea level, partial basin restriction may have occurred, allowing local freshwater inputs to lower $\delta^{18}O_{sw}$ values. In maritime regions such as modern Bermuda, despite lack of elevation or substantial landmass, precipitation and precipitation-derived runoff is depleted in $\delta^{18}O$ by 4–5 ‰ VSMOW relative to seawater (Zhang et

al., 2021). Additionally, surface runoff may contain dissolved terrestrial organic matter depleted in $\delta^{13}C$, which could explain the correlation between $\delta^{18}O_{sw}$ and $\delta^{13}C$ through our study period. However, it is difficult to explain how local freshwater in a subtropical environment surrounded by low-lying landmasses could be substantially colder than local seawater, so we favor the variable water mass hypothesis over a local freshwater contribution.

Yet another possible interpretation of the co-variation between fluctuating temperature and $\delta^{18}O_{sw}$ trends is via climate-related reorganization of atmospheric circulation, moving broad precipitation bands north and south over the study region. An enhanced hydrological cycle has been hypothesized for the latest Cretaceous (e.g., Woelders et al., 2017) due to increased zonal surface winds and ~20 % stronger Hadley circulation in the Northern Hemisphere (Bush and Philander, 1997). If, during

warmer climates, precipitation bands moved such that the Maastrichtian type region experienced less rainfall, $\delta^{18}O$ enrichment of surface waters would coincide with warmer temperatures and vice versa. This is difficult to assess, even in global climate model simulations, as model proficiency in reconstructing paleo-precipitation regimes is difficult to validate/calibrate. The rough correlation with $\delta^{13}C$ is also difficult to explain in this scenario.

## 6 Conclusions

This study presents a new clumped isotope-based paleotemperature time series from the latest Cretaceous in northwestern Europe. The $\Delta_{47}$ analyses from the Maastrichtian type section and surrounding outcrops reveal a mean temperature of $20.4 \pm 3.8$ °C and an average $\delta^{18}O_{sw}$ of $+0.2 \pm 0.8$ ‰ VSMOW, consistent with a subtropical shallow marine environment. This average temperature aligns well with other paleothermometry studies from similar paleolatitudes and with modeled Maastrichtian paleotemperatures. Increasing $\Delta_{47}$-based temperatures at ~66.4 Ma, in conjunction with other indicators, are interpreted to reflect a regional manifestation of the global LMWE and directly link pre-KPB warming in northwest Europe to $CO_2$ emissions from Deccan Traps volcanism. Simultaneous increases in temperature and $\delta^{18}O_{sw}$ during the LMWE mask the timing of warming in $\delta^{18}O_{carb}$-based temperature reconstructions, which may explain a more delayed apparent timing of the LMWE in other studies. Seen now in two spatially-distant records, this deserves further exploration and emphasizes the need for using paleothermometry tools that take into account variations in $\delta^{18}O_{sw}$.

Covariations in temperature, $\delta^{18}O_{sw}$, and (to a lesser degree) $\delta^{13}C$ from horizon to horizon may reflect the varying influences of different water mass at different times. We hypothesize that colder waters with lower $\delta^{18}O_{sw}$ and $\delta^{13}C_{DIC}$ may have originated from the Arctic and reached the study site via a passageway between Greenland and Scandinavia (Fig. 1a) during periods of lower sea level when emergent landmasses to the south acted as barriers. Warmer waters with higher $\delta^{18}O_{sw}$ and $\delta^{13}C_{DIC}$ may have been sourced from a proto-Tethys to the southeast or proto-Atlantic to the southwest (Fig. 1a) and flushed the study region during intervals of higher sea level.

Overall, the fossils of the Maastricht Formation are effective paleotemperature archives that reveal linkages between local sea level changes, Deccan Traps volcanism, and ocean temperature on the European continent in the lead-up to the end-Cretaceous.

## Data availability

All data presented in this paper are available in the EarthChem database at https://doi.org/10.26022/IEDA/112046.
Further data are available in the Supplement:

- **Figure S1:** oxygen isotope values of shell specimens and bulk carbonate;
- **Figure S2:** carbon isotope values of shell specimens and bulk carbonate;
- **Figure S3:** carbon isotopic composition vs. oxygen isotopic composition in well-preserved shells and bulk matrix;

▪ **Figure S4:** reconstructed seawater oxygen isotopic composition through time;

     ▪ **Figure S5:** covariation of oxygen isotopic composition of carbonate, water, and temperature;

     ▪ **Table S1:** published paleotemperature data for other studies reconstructing Maastrichtian marine temperatures;

     ▪ **Table S2:** age model of Vellekoop et al. (2022);

     ▪ **Table S3:** preservation data for all samples, including SEM preservation determinations and trace element
concentrations;

     ▪ **Table S4:** summary of in-house carbonate standards.

## Author contribution

SVP, JV, and HEO designed the study. SVP and JV collected the samples. HEO, SVP, MMJ, and SRS measured and analysed the data. HEO and SVP prepared the manuscript, with contributions from all co-authors.

## Competing interests

The authors declare that they have no conflicts of interest.

## Acknowledgements

We thank Ashling Neary (SCIPP lab), Angela Dial (MEAL), Lora Wingate and Kacey Lohmann (SIL) for assisting in sample preparation and measurements at the University of Michigan. We also thank UM's Department of Earth and Environmental
Sciences for the Turner Research Award and the Geologic Society of America for the Graduate Student Research Grant, both of which helped to fund this project. This work was also funded by grant 12Z6621N of the Research Foundation Flanders (FWO) to JV. The GSG sample was collected as part of a field trip related to the International Conference on Paleoceanography in 2014 and we thank the organizers and co-participants.

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
