# Peer review of "Clumped-isotope-derived climate trends leading up to the end-Cretaceous mass extinction in northwest Europe"

_Climate of the Past, 2021_

## Author Comment (AC2)

| Standard | Kiel | MAT 253 | | Nu Perspective | |
|---|---|---|---|---|---|
| | Calibration (2014, 2019) | Long Term (2015, 2018–19) | These Sessions (2016–17, 2019–20) | Long Term (Jul–Dec 2020) | These Sessions (2021A, 2022A) |
| **CM** | (N=17) | (N=27) | (N=18) | (N=77) | (N=18) |
| $\Delta_{47\text{-CDES25}}$ | | $0.396 \pm 0.024$ | $0.398 \pm 0.023$ | $0.4066 \pm 0.018$ | $0.395 \pm 0.018$ |
| $\delta^{18}O$ | $-2.16 \pm 0.09$ | $-2.05 \pm 0.29$ | $-2.03 \pm 0.16$ | $-2.16 \pm 0.07$ | $-2.21 \pm 0.13$ |
| $\delta^{13}C$ | $2.05 \pm 0.04$ | $1.89 \pm 0.09$ | $1.92 \pm 0.05$ | $2.02 \pm 0.021$ | $2.03 \pm 0.05$ |
| **OO\*** | (N=8) | (N=23) | (N=15) | (N=35) | (N=18) |
| $\Delta_{47\text{-CDES25}}$ | | $0.683 \pm 0.022$ | $0.700 \pm 0.029$ | $0.6864 \pm 0.015$ | $0.679 \pm 0.029$ |
| $\delta^{18}O$ | $-0.16 \pm 0.09$ | $-0.06 \pm 0.18$ | $0.23 \pm 0.16$ | $-0.02 \pm 0.07$ | $0.05 \pm 0.17$ |
| $\delta^{13}C$ | $4.77 \pm 0.09$ | $4.72 \pm 0.09$ | $4.76 \pm 0.08$ | $4.87 \pm 0.03$ | $4.89 \pm 0.06$ |
| **CORS** | (N=5) | (N=12) | (N=7) | (N=38) | (N=3) |
| $\Delta_{47\text{-CDES25}}$ | | $0.723 \pm 0.014$ | $0.713 \pm 0.045$ | $0.7243 \pm 0.016$ | $0.715 \pm 0.011$ |
| $\delta^{18}O$ | $-4.07 \pm 0.08$ | $-4.10 \pm 0.14$ | $-3.84 \pm 0.11$ | $-4.02 \pm 0.06$ | $-4.11 \pm 0.07$ |
| $\delta^{13}C$ | $-3.64 \pm 0.09$ | $-3.70 \pm 0.06$ | $-3.68 \pm 0.09$ | $-3.64 \pm 0.04$ | $-3.62 \pm 0.03$ |
| **Pica** | | | | (N=67) | (N=15) |
| $\Delta_{47\text{-CDES25}}$ | | | | $0.7072 \pm 0.023$ | $0.701 \pm 0.014$ |
| $\delta^{18}O$ | | | | $0.52 \pm 0.06$ | $0.44 \pm 0.10$ |
| $\delta^{13}C$ | | | | $3.35 \pm 0.03$ | $3.33 \pm 0.05$ |
| **Ice** | | | | (N =30) | (N=20) |
| $\Delta_{47\text{-CDES25}}$ | | | | $0.7409 \pm 0.011$ | $0.727 \pm 0.019$ |
| $\delta^{18}O$ | | | | $2.08 \pm 0.08$ | $2.04 \pm 0.15$ |
| $\delta^{13}C$ | | | | $2.19 \pm 0.03$ | $2.20 \pm 0.06$ |

Table summarizing in-house standards run over many years at the University of Michigan, demonstrating reproducibility on multiple machines and consistency through time and across machines.

---

## Author Comment (AC3)

Figure showing SEM preservation (and associated trace element concentrations) of 9 samples from the study. Any sample determined to fail either trace element of SEM screening would not be included in further interpretations.

[Figure]

A figure comparing our temperature time series determined by D47 measurements (blue) to the composite time series of Hull et al. (2020) (grey points and green solid line). Also shown is the time series that would have been interpreted from our d18Ocarb alone, assuming a constant d18Oseawater, as is done in many records included in Hull et al. In this case, the peak temperatures implied by d18Ocarb occur later than the peak temperatures implied by the D47 results.

---

## Author Comment (AC5)

Figure showing clumped-isotope-based temperatures for all samples. Red samples have not passed screening and were not interpreted further, although the close agreement between these temperatures and those of other well-preserved samples from the same horizons suggests our screening criteria are quite conservative. Compared to the older version of this figure, a few new points have been added, error bars have been added in the x-direction to show uncertainty in age model at different horizons. Uncertainty in the timing of the onset of Deccan volcanism has been indicated by grey shading. This date represents the oldest radiometric date of Deccan lavas (Sprain et al., 2019)

---

## Author Response (AR1)

RESPONSE TO REVIEWERS AND LIST OF CHANGES

**LIST OF MAJOR CHANGES**

In response to reviewer comments and in an effort to strengthen and justify the conclusions of the manuscript as much as possible, we have made the following changes.

1. We have **revised our screening techniques** to separately rate dissolution and presence of secondary diagenetic calcite growth, as the later influences the isotopic composition whereas the former does not (or does to a much lesser degree). We have included significantly more description of this process, including a **new main text figure with nine SEM images** and a supplementary table dedicated to screening criteria.

2. We have **analyzed additional samples for clumped isotopes** to increase the number of points per horizon and strengthen conclusions about temporal trends. A third sample from the basal Nekum Member agrees with two previous samples in documenting a clear warming signal which we infer to be the Late Maastrichtian Warming Event. In the Meerssen Member, the new sample is warmer than a previous single cool sample, generally reducing or erasing an apparent cooling trend up to the K-Pg boundary. We have changed the text and figures in many places in relation to these new data, number of samples, horizon and site averages, etc. Isotopic sample means can now be found in a **new main file data table**.

3. We have **reduced the emphasis on reprocessing the Meyer et al 2018 and 2019 datasets**, as reprocessed data using Brand/IUPAC parameters was actually already published in the supplementary material of the original papers and is therefore not a new contribution. This data is still included as a point of comparison, but we no longer claim to have done the reprocessing ourselves. We remove the table focusing on this data from the main text, although these samples are still included in Supplementary Table S1.

4. We have made a **new final figure to focus on the temporal temperature trend through the Late Maastichtian Warm Event** and compare our data to a composite temperature record recently published by Hull et al. 2020 instead of focusing on a latitudinal temperature gradient. As we are only reporting temperatures from a single site, this seemed more appropriate. We still compare to published data from other latitudes in the text, but no longer show this comparison in a figure. We have accompanied this new time-series figure with a few new paragraphs of discussion.

5. We have **added significantly more details to the methods section**, as this will now likely be the first paper from our group using our new mass spectrometer and sample preparation set up. We have also included a new supplementary table documenting performance of in-house carbonate standards through time and across our two mass specs, demonstrating clearly that data collected over many years and across two machines are comparable and combinable into a single dataset.

Below are the point-by-point responses to reviewers. These are essentially the same as what was posted to the online open comments, with a bit more detail in some places and less detail in others where we instead refer you to the revised manuscript.

**REVIEWER #1**

**Q: O'Hora et al., provide clumped isotopic data to reconstruct seawater temperatures and δ¹⁸O from fossil bivalves from the Cretaceous section of the Maastrichtian. This new record helps to understand 1) the regional and global climate response to the volcanism of the Deccan traps and 2) the changes in ocean circulation with alternating southern and the Arctic Ocean waters. This dataset and article have publication potential. However, before publishing it, there are some points that need to be improved.**
A: Thank you for your comments, Reviewer #1. We have addressed them to the best of our ability and describe responses below.

**Q: The method part about clumped isotopes should be detailed. It would be good to detail how you processed your data and how you combined the two datasets (Thermo and Nu). It looks like you used ETH standards, however, with the Thermo you didn't measure ETH.**
A: As you point out, our early clumped isotope data was collected on one machine (MAT 253) and corrected using a gas-based reference frame and our later data was collected on another instrument (Nu Perspective) and corrected into the I-CDES carbonate-based reference frame. For some samples not analyzed for their clumped isotopic composition, we measured just d13C and d18O using a MAT253 + Kiel IV device. To ensure consistency, alongside samples we analyzed three to five in-house carbonate standards, whose isotopic compositions were tracked through time and across machines. We have compiled a table with summary statistics on these in-house standards through time and on both machines that has been added to the supplementary material of our manuscript (Supplementary Table S4). This table shows that d13C, d18O, and D47 values of these standards have been constant (within error) through time and across machines, so all sample data can confidently be combined and treated as a single dataset.

**Q: How reproducible is your data?**
A: General reproducibility can be demonstrated by the long-term standard deviation of the above-mentioned in-house carbonate standards which are measured frequently yet not used in the absolute reference frame correction itself. For the MAT253 analyses, long-term 1sd on D47 is 0.020‰. For the Nu Perspective, long-term 1sd on D47 is 0.018‰. Long-term 1sd for d13C and d18O are generally better than 0.1 and 0. ‰, respectively. This is now shown in Supplementary Table S4.

**Q: Are your ETH standards values comparable to Intercarb values?**
A: Yes, when ETH standards are used for correction to the I-CDES, we assign values from Bernasconi et al 2021, so they are by definition extremely comparable to identical to Intercarb values. ETH standards were not measured on the MAT253 when a gas-based reference frame was used.

**Q: Is this the first article published with data from your Nu-perspective? If no, please cite the article that describes the procedure, if yes, please give more details on the methodology of this MS.**

A: Yes it is, although we did not know it at the time of original submission. We have added a large amount of detail on the preparation (see section 3.3), analysis, and data processing methods, along with citing Mackey et al (2020) which describes details of a similar instrument used in a slightly different mode (coldfinger mode).

**Q: How did you convert the data from the D47 into an absolute repository (software, codes)?**

A: Between 2016-2020, a first set of fossil shell powders were measured for $\Delta_{47}$ using a Thermo-Finnigan MAT 253 dual inlet isotope ratio mass spectrometer and a manual sample preparation device described in detail by Defliese et al. (2015). These samples were converted into the absolute reference frame using gas standards and long, fixed windows. In 2021 and 2022, additional powders were analyzed in the University of Michigan SCIPP lab using a Nu Perspective isotope dual inlet ratio mass spectrometer connected to a NuCarb automated sample preparation device. Isotopic values were converted into the Intercarb Carbon Dioxide Equilibrium Scale (I-CDES25) absolute reference frame using $\Delta_{47}$ values for four ETH standards defined by the Intercarb project (Bernasconi et al., 2021) and long, fixed windows. A longer, more detailed description has been added to the methods section of the paper (section 3.3) and all calculated equilibrium gas line slopes and transfer function slopes and intercepts are included in the final data table archived in EarthChem ClumpDB.

**Q: You can also specify the number of measurement sessions and how you calculate your temperature uncertainties. It must be written in the section method, not in the caption of a figure. In the legend, the definition of your uncertainties is not clear.**

A: There is a blur between what counts as a "measurement session" (which might be a few months long between major maintenance or power outages) and a "correction window" (a period from which standards are selected to correct a subset of data). Multiple correction windows make up one measurement session. We now describe the measurement sessions in the methods section (section 3.3), along with significantly more detail about data treatment and calculation of uncertainties. For information about specific correction windows (which we feel is too granular for most readers), someone can easily find this information in a column labeled "window" in the EarthChem data template.

**Q: Also, some citations are missing such as Brand et al., 2010 and Daeron et al., 2017 for the 17O corrections and Meckler et al., 2014 for the ETH standards.**

A: Added many of these and others where appropriate.

**Q: Would it be possible to replicate some samples to reduce the uncertainties on the temperatures?**

A: Since receiving this review, we have added additional replicates to the few samples with the worst reproducibility. We have also done more thorough replicate screening (description of which can be added to methods), removing a single replicate if it falls more than 2.5 standard deviations from the mean of the remaining replicates. These combined efforts improved error bars on a number of samples. Additionally, we have now analyzed a few more new samples to improve the number of samples per horizon and to make our conclusions more robust. These new data reinforce prior conclusions, such as the timing and magnitude of warming during the Late Maastrichtian Warming Event.

**Q: You have measured different carbonates (matrix and fossil bivalves; different species). However, you are not comparing this data. For example, how do you explain the temperature difference between Acutostrea and Agerostrea at 66.5 Ma? A discussion of this comparison would improve the manuscript.**

A: We disagree with the reviewer that the temperature difference between Acutostrea and Agerostrea at 66.5 Ma is meaningful, as they are comparing two single samples from one horizon. Yes, that is the horizon with the largest variability, and it happens to include multiple taxa, but we see a similar level if variability between samples of the same taxa from the same horizon. Fossils found at the same horizon were not necessarily living at exactly the same time, so may be accurately representing their living environment but the living environment is not exactly the same as for the other shell from that horizon. Also, for the most part, shells included in this study are small oysters, so it is possible that our drilled powder has aliased certain seasons in one shell vs. another.

**Q: In general, I would suggest referring more often to figures and tables.**

A: More references to figures and tables were added throughout the text.

**Q: Did you put your uncertainties at 1 or 2 sigma in the figures?**

A: Errors were 1 sigma external SE. This has been added to the caption of many figures and explained more fully in the methods section (section 3.3).

**Q: You talk a lot about SEM images; however, no image is shown. Please put at least 3 with original material, traces of dissolution and secondary growth. You can include them in Figure 3.**

A: We have added a figure (new Figure 3) showing SEM images of nine shells showing different levels of preservation, along with their trace element concentrations and our overall preservation rating and include much more detail in the methods section about how these ratings were carried out and which samples were excluded as a result.

**Q: I would suggest removing Figure 4 (as you have the same one in Figure 5) or skipping Figure S3 in the main text, which is much more detailed.**

A: We think having a zoomed-in version of the temperature profile is beneficial to see interspecies differences that are not visible in the big composite figure (previously Figure 5, now Figure 6), but we agree that Figure 4 and S3 are redundant. We now prefer to include Figure S3 in the main text (new Figure 5) which shows the removed data points, as well as the species-level differences.

**Q: A table with the data, such as the last table in the material supplement, can be included in the main text.**
A: We agree and have now added a summary table of all isotopic compositions to the main text (Table 1). We think the trace element concentrations and SEM indices can remain in the supplement in a second table (Supplementary Table S3). Combining them into a huge table in the main text just leads to it being harder to read.

**Q: In figure 5 you can label each graphic with a letter (a, b, c…) that would help to read the caption.**
A: Done.

**Q: I would suggest adding a figure to illustrate the comparison between your data and previously published data.**
A: We have added a new figure to the main text, comparing our data to the composite dataset of Hull et al., 2020, which combines all published records over the Late Maastrichtian Warm Event. This now replaces the latitudinal gradient figure, as the temporal variation in temperatures is in better alignment with the emphasis of this paper on the LMWE.

**Q: This [discussion] section can be more detailed. You have mixed in the same paragraph the comparison of your data with today's climatic conditions and with previously published data. I would suggest adding more details, splitting the paragraph into 2 parts and also changing the name of the section.**
A: We have combined this section with the following one, and rearranged the presentation of ideas.

**Q: I am not sure you can cite an article that is in preparation.**
A: This paper is now accepted, so we continue to cite it, now without issue.

**Q: Lines 79-80: Reference?**
A: Added.

**Q: Fig 2 the legend in the square can be larger**
A: Done.

**Q: Lines 124-125: add the reference of Huyghe et al., 2021**
A: Done, plus added more discussion about vital effects here.

**Q: Lines 123-130: it seems that you mixed up methodology and material**
A: Rearranged.

**Q: Lines 216-218: specify which samples are not included in the discussion**
A: Added much more detail to this, including information on sample screening in the new main text table.

**Q: Line 222: change ~ 20$^O$C to the real values X $^O$C +/- X $^O$C**
A: Done.

**Q: Lines 260-262: I would suggest to put more explanation**
A: Elaborated here more.

**Q: Section 5. Change the name to Discussion**
A: Done. Good catch.

**Q: Lines 384-385: I would suggest moving this conclusion at the end of the next paragraph**
A: We did some rearrangement in this section which hopefully addresses this issue.

**Q: Lines 408: what is the other evidence?**
A: The ones listed in the following paragraph. Reworded to make this clear.

**Q: Line 419: what are the other records? On the basis of which proxy?**
A: In light of our new additional samples, we have removed this portion.

**Q: References: Add et al., for Petersen et al., 2019, like you did for intercarb's article**
A: Done.

**Q: Data file in EarthChem: The names of the working sample are sometime different from the Replicate ID.**
A: A new version of the EarthChem file will be uploaded with this resubmission to both fix these mentioned issues and include newly analyzed sample and standard replicates.

**Q: Table S2: add the number of replicates**
A: This has now been added, although the total table is split now into Table 1 (isotopes and summary of screening) and Supplementary Table S3 (details of screening)

**REVIEWER #2**
Q: Firstly, I must sincerely apologize for the lateness of this review.  Despite being vaccinated and having received a booster, I still managed to get CoVid and was sick for weeks.  I deeply apologize to the authors for this unreasonable delay in evaluating their paper. The authors report on a very nice record of temperature change from sections in the Netherlands and Belgium following the eruption of the main phase of the Deccan traps.  They find an apparent spike in temperatures around 8*C above background, followed by a slow decline over the following few hundred thousand years.  This is a relatively straightforward paper with what could be considered conservative interpretations of some really quite provocative data.
A: We are sorry to hear about your health struggles and appreciate you getting around to our review in the end. Thank you for the kind words about the provocativeness of our data.

Q: What these data indicate is that even though the temperature change due to the eruption of the Deccan traps was apparently significant (ca. 8*), it was also relatively short lived, having been mostly erased in 75-100kyr. This is consistent with the nature of the $pCO_2$ signal we see from other LIPs such as the central Atlantic magmatic province ($CO_2$ blips can result in a doubling, but are mostly erased over the subsequent 2-300kyr (Schaller et al 2012)).  These temperature estimates are also consistent with earlier estimates (eg wilf 2003, woelders et al 2017, zhang et al 2018), although the authors actually show a much cleaner signal than some of the previous work (In my opinion). Given that, the Deccan warming and the K-Pg extinction strata are pretty clearly unrelated events (as is the prevailing thinking). BUT it also makes me wonder, what are the true effects of this warming?  It is very short, basically a blip with a duration of no more than 100kyr, because the samples at ~66.3 Ma are indistinguishable from background.  In fact, it is possible it is much shorter than 100kyr because of the limits of the sampling resolution here.  However, what I see in the data is a longer-term _cooling_ from 66.37 to 66.07 Ma.  In fact, this is the most significant secular signal in the data – what does this mean?  It's of the right timescale to be the predicted $pCO_2$/temperature decrease due to continental weathering following the emplacement of the Deccan traps (e.g., Desert et al 2001, 2003).  I think the authors should highlight this!
A: We appreciate this suggestion and now mention of silicate weathering feedbacks and timescales, with some of the citations you mention. However, the secular trend you picked up on does not hold up so well with the addition of a few additional samples analyzed recently as part of this revision. In particular, in the Meerssen Member we previously only had one sample which was particularly cold and set a lot of the cooling trend you described. Now we have a second sample from that horizon which is closer to the "baseline" temperature of ~20°C, so the sustained cooling trend has largely disappeared. It remains the case, however, that the clear warming shortly after the Deccan Traps onset is gone after ~100kyr or less, and this finding has only become stronger with the addition of more data points.

Q: There are always questions of timescale, so just so it's clear (and I probably missed it), but has the ejecta horizon/dust/Ir level etc. been positively identified in these N. European sections?

A: While an ejecta blanket was not preserved in the very shallow marine setting (i.e above fair-weather wave base) of the type-Maastrichtian region, the K-Pg boundary is positively identified (see Smit & Brinkhuis 1996; Vellekoop et al. 2020, https://doi.org/10.1111/pala.12462). The typical planktonic foraminiferal 'disaster' assemblage (Smit & Zachariasse 1996), the presence of the earliest Paleocene dinocyst marker taxon Senoniasphaera inornata (see Brinkhuis & Schiøler 1996; Herngreen et al. 1998) and 87Sr/86Sr analyses of well-preserved foraminifera from the clay layers of the Geulhemmerberg underground galleries (Vonhof & Smit 1996) have all demonstrated that the K-Pg boundary occurs at the base of unit IVf-7 of the Meerssen Member, at the base of a sequence of shell hashes and clay layers. These details have now been added to section 2.

Q: Can we be certain that the horizons identified as coincident with the initial phase of volcanism are indeed correctly correlated? Bulk C-isotope data (Line 191) does not instill a lot of confidence…
A: The identification of the Deccan Traps interval is not only based on bulk d13C data. As highlighted in Vellekoop et al. 2022 (Newsletters on Stratigraphy), the age model for this succession is based on a combination of biostratigraphic markers (ammonites, belemnites, dinocyst) and the isotope record. Moreover, the presence of the acme of the dinocyst marker taxon Palynodium grallator, a marker for the LMWE (Vellekoop et al,. 2019, https://doi.org/10.5194/bg-16-4201-2019) in the type-Maastrichtian record (Schioler et al., 1997) is clear evidence for this phase of Deccan Traps volcanism. Greater details on stratigraphy have been added to section 2.

Q: If I were to be critical of one point of this paper, it is the confidence with which the correlation is made between marine sections in Europe and the terrestrial record of Deccan volcanism that relies mostly on radiometric ages from India. This should be expanded upon and the tie points made more explicit (is the age model published?).
A: Radiometric dating places the large outpouring phases of the Thakaurvadi to Puladpur lava deposits in the time interval between 66.3 and 66.05 Ma (e.g. Schoene et al. 2019, Science). At this point, it is well established that the Late Maastrichtian Warming Event is related to these Deccan outpouring phases, for example also highlighted in the recent paper by Nava et al. in PNAS (https://doi.org/10.1073/pnas.2007797118). Hence, no direct correlation with any terrestrial records in India is required. The age-model for type Maastrichtian (based on a combination of biostratigraphy and chemostratigraphy described above) clearly indicates that the Nekum & Meerssen members fall within the LMWE interval. There is even a specific marker (the dinocyst *Palynodinium grallator*) for this warming event present in these records. We hope any concerns considering the age model of the type-Maastrichtian record are waylaid by the publication of the new age model for this section by Vellekoop et al (2022).

To be more transparent about the uncertainty in our age estimates and correlation, we have added uncertainty in the age model as a horizontal error bar in Figure 5 showing Temperature vs. Age, as well as uncertainty in the onset of Deccan volcanism (grey shading). The onset of Deccan volcanism here is pinpointed as the oldest date (with accompanying error) coming from Deccan lavas (Sprain et al., 2019), and thus may be an underestimation of the onset.

**Q: Can you add the timing (duration?) of Deccan volcanism to Fig. 5? I need some frame of reference for where I should be looking for the increase in temperature. The line at 66.4 is helpful but what is the duration?**

A: We have thickened the bar showing the onset, now with error. This point is defined by the oldest dated volcanic rock from India (Sprain et al 2019). In terms of when to expect warming…$CO_2$ driven warming does not happen instantaneously, and the $CO_2$ addition would be expected to continue (perhaps nonlinearly) throughout the full period of eruptions, which extend well past the KPB. In other records showing the LMWE (e.g. Petersen et al., 2016, nature communications), there is a strong warming shortly after onset of volcanism.

**Q: It may just be my general skepticism of absolute sea level estimates in ancient rocks, but can the authors expand upon the goals of the sea level reconstruction here? To what end?**

A: This is a sea level reconstruction from another paper (Schioler 1997). We put it here as a possible explanation for our co-evolving temperature and d18Osw records. We propose that changes in sea level correlate with our study site experiencing differing water masses due to the paleogeography.

---

## Author Response (AR2)

We have responded to the very minor remaining suggestions of the reviewers as follows:
*Note: their line #s do not align with the submitted version, but we found mentioned sections as best we could from other clues in each comment.*

- Figure 1: it was not obvious that panel C is a zoom of B without reading the caption.
    - **Lines added to show the zoom in.**

- Line 361: define what a good replicate is. The explanations come too late in the text.
    - **Moved up key sentences.**

- Line 425: Is the acid fractionation for oxygen or for the clumped? According with InterCarb recommendations, no acid fractionation factor is required for clumped.
    - **Added sentence: Although use of ETH standards does not require an acid fractionation factor when applied to an entire dataset, to combine data from gas-based and carbonate-based reference frames requires normalization to the same acid digestion temperature.**

- Lines 433-440: it is not clear; can you rephrase a bit, please. For example, line 434, what are "these 3 standards"? And line 438: your table is in supplementary material; it would be good to remember which are the 3 instruments.
    - **"Three" was a carry over from a previous version. Now says "for in-house carbonate standards". Added "on the same Kiel IV + MAT253 used for bulk sample $\delta^{13}C$ and $\delta^{18}O$"**

- Line 448: move the sentence about Anderson et al., 2021 before starting to discuss d18Osw.
    - **Done**

- Line 472: repetition of long-term SD/reproducibility, already written earlier.
    - **Not repetitive. We are describing something different here, although similar.**

- Line 472: what is the SEM preservation rate? Do you have a reference?
    - **Defined here, in later paragraphs. Added "see below".**

- Lines 530-538: repetitions
The rest of the paragraph is more methodology, I would suggest moving it in methodology part.
    - **Done.**

- Figure 3: the thresholds for trace element concentration as diagenesis proxy are not clearly explained in the text.
    - **Many citations are given.**

- Lines 541-542 may be moved to methodology also and a ref is missing.
    - **Reorganized this section.**

- Line 708: only differed by 0.1-0.2 per mill compared to?
    - **Compared to kiel-derived values. Added text.**

- Line 713: Fig. S1- I couldn't see the outliers, can you highlight them please? Can you also remember why you exclude them?

- o **Label of "outlier" here does not mean a bad point, just one that deviates clearly from the rest of the group. We removed this terminology and instead calculated section average using all points which only changed things by 0.1 permil.**

- Line 713-715: It may be more accurate to present the data as a range like you did for the bulk data.
  Same comment for the d13C and Figure S2.
  - o **Same as above**

- Line 735: add the SD after "within the error of each other".
  - o **Can't do this because different errors on different samples based on reproducibility. We refer the reader to the error bars on the figure.**

- Regarding the discussion of the vital effect. I am still not convinced by the error bar argument. Your uncertainties are quite large at 1 SD, and the difference in absolute values can be up to 5 deg.C (for the sample at around 66.5 Ma). I advise being more nuanced by explaining that your data tends to support past observations on the absence of vital effects, since the values are within error bars. However, more replicates are needed to reduce the error bars in order to be able to confirm the absence of vital effects.
  I really appreciate the end of the discussion about seasonality and age of living biases. I would suggest to also extend this discussion to the sample at 66.1 Ma.
  - o **Added: although further replication to reduce error bar size would be necessary to rule out small vital effects**
  - o **Extended the argument as suggested.**

- Figure 6: I would suggest being more specific when writing "see text", by mentioning the section you are referring to.
  - o **We meant section 3.3, now added.**

- Line 855: please check the text to capitalize SD (idem line 898 for example)
  - o **Done.**

- Line 906: add 1SD
  - o **Done**

- Line 975: What is the uncertainty about your age model? Can it be used as an additional argument for the good agreement between the beginning of the eruptions and the warmer T-D47?
  - o **Uncertainties are already stated in the preceding paragraph. We have added them to Figure 7 to highlight the overlap between the onset of Deccan volcanism and warming temperatures in our record.**